# Global Linear Convergence of Inexact TD Under Generalized Smoothness

## Abstract

Recent work has analyzed temporal-difference (TD) learning with target networks through an optimization view and established linear convergence under a force-dominance condition, but these results typically rely on global smoothness, i.e., a uniform upper bound on curvature. This assumption can fail even when the inner problem is well posed, since curvature encountered during training can grow with the scale of TD-residual-induced gradients. We retain the stabilized regime in which the inner problem is strongly convex in the optimization variable, in order to isolate upper-curvature growth effects. Under generalized smoothness, where the Hessian norm may grow with gradient scale via a nondecreasing profile $\ell(\cdot)$, we analyze the inexact TD recursion with $K$ inner gradient steps per target refresh and propose a curvature-checked constant stepsize rule that ensures global stability without requiring a global smoothness constant. Our main result proves global linear convergence under force dominance with a single trajectory-dependent admissibility requirement governed by the maximum gradient magnitude $M$ encountered along the run. This yields an explicit scaling law: the largest admissible constant stepsize decays as $1/\ell(cM)$, for a universal constant $c$, and maintaining a fixed contraction requires $K$ to grow proportionally to $\ell(cM)$. In the special case of uniformly bounded curvature, our result reduces to the classical global-smoothness regime; under curvature growth, the worst trajectory gradient scale controls both stability and attainable convergence speed, yielding a mechanism-level interpretation of why curvature-aware step control can matter in stabilized TD-style optimization.

## 1 Introduction

Temporal-difference (TD) learning is widely regarded as a central algorithm in reinforcement learning (RL), and a large fraction of modern RL methods build directly on the core ideas it embodies. TD has played a key role in several landmark RL successes (Mnih et al., 2015; Silver et al., 2016), and its influence extends beyond RL itself to other areas of AI, as well as to fields such as economics and neuroscience. At the same time, TD is known to exhibit divergence in carefully constructed examples (Tsitsiklis & Van Roy, 1996; Baird, 1995; Sutton et al., 1998). Empirically, however, such divergence is uncommon even when TD is paired with highly expressive function approximators. This tension motivates a more refined theoretical understanding of TD and a strengthening of existing convergence results in directions that better reflect practice.

In this paper, we adopt this optimization perspective, framing TD as a two-level iterative optimization algorithm (Asadi et al., 2023b) that proceeds as follows:

$$\theta_{t+1}{}^1 \approx \arg\min_w H(\theta_t, w). \tag{1}$$

The objective $H$ is obtained by (a) selecting a function approximator (e.g., a neural network) and (b) specifying a discrepancy between successive predictions (e.g., a squared-error criterion). We provide additional examples later.

---

[1] While we adopt $\theta$ to denote the target variable, consistent with recent optimization-centric RL literature (Asadi et al., 2023b), we wish to clarify for the broader RL community that this variable plays the functional role of the traditional target network weights, often denoted as $\theta^-$.

The formulation in (1) highlights two distinct parameter roles. The target parameter $\theta_t$ is held fixed at iteration $t$, while the optimization parameter $w$ is updated to reduce the associated loss. In deep-RL terminology, $\theta_t$ corresponds to the target-network parameters, whereas $w$ corresponds to the online-network parameters. A broad class of RL algorithms can be cast in this template, differing primarily in how the minimization is carried out in practice. At one end, the original TD method (Sutton, 1988) takes a single gradient step in $w$ and then updates $\theta$. A commonly used intermediate regime in deep RL performs $K$ gradient-based updates of $w$ while freezing $\theta$ (Mnih et al., 2015). At the other end, Fitted Value Iteration (Gordon, 1995) effectively solves each inner problem exactly when closed-form solutions are available. Accordingly, understanding the iterative optimization mechanism in (1) helps clarify TD itself and the behavior of many TD-style algorithms.

The optimization process (1) has been extensively analyzed in the classical linear-function-approximation setting with a squared-error objective. In particular, when $K = 1$ (TD without a frozen target network), the seminal analysis of Tsitsiklis & Van Roy (1996) established convergence guarantees. Building on this line of work, Lee & He (2019) investigated the more general regime $K \geq 1$. Their analysis treats gradient descent with fixed $K$ as an approximate inner solve, which induces an approximation error at each outer iteration; this per-iteration error accumulates over time and therefore appears explicitly in the final guarantee. As a consequence, (Lee & He, 2019) do not obtain convergence exactly to the TD fixed point, but rather to a neighborhood whose size is governed by the accumulated approximation error.

More recently, Asadi et al. (2023b) studied this optimization view of TD and derived sharp stability conditions that can be interpreted through a *force-dominance* lens: convergence requires the optimization force (curvature of the objective in the inner variable $w$) to strictly dominate the target force (the sensitivity of the induced target mapping). While such results yield fixed-point convergence for arbitrary $K \geq 1$, their guarantees hinge on a strong smoothness requirement, namely global $L$-smoothness. Specifically, they assume the gradient of the inner objective is uniformly Lipschitz continuous in order to certify descent for the inner optimization steps.

As motivation, modern deep RL with neural critics and value heads can exhibit pronounced, highly non-uniform upper curvature in the associated optimization landscapes. A growing body of evidence in deep learning shows that training trajectories routinely encounter phases of *progressive sharpening* and operate near an *edge-of-stability* threshold, where the largest Hessian eigenvalues track the stepsize scale rather than remaining uniformly bounded by a global constant (Cohen et al., 2021; Gilmer et al., 2022; Damian et al., 2023; Roulet et al., 2024; Long & Bartlett, 2024). In such regimes, a single global $L$ controlling $\|\nabla_w^2 H(\theta, w)\|_{\mathrm{op}}$ may be unavailable or overly conservative as a descriptor of the locally stable stepsizes encountered along training trajectories, even for stabilized or regularized inner objectives. TD further amplifies this issue because the optimization problem is inherently nonstationary: in the optimization view, the function minimized at iteration $t$ changes across outer updates since the bootstrapped targets depend on the current (or lagged) value parameters, as in target-network constructions (Mnih et al., 2015; Lee & He, 2019; Bengio et al., 2020; Asadi et al., 2023b). Empirically, this moving-target nonstationarity can destabilize gradient-based training and is associated with progressive sharpening of the loss landscape in deep RL (Lyle et al., 2023).

This motivates a convergence theory compatible with *unbounded* and *state-dependent* upper curvature in the *inner* TD objective, without requiring globally upper bounded Hessians. At the same time, analyzing fully coupled deep-TD dynamics with nonconvex function approximation is a separate challenge and is not the obstacle we target here. Instead, following the optimization-theoretic TD formulation (Asadi et al., 2023b), we work in the stabilized regime assumed by prior optimization viewpoints: for each fixed target $\theta$, the inner problem in $w$ is well-posed and strongly convex (Asadi et al., 2023b), yielding a unique minimizer map and enabling a clean contraction-based stability argument. This modeling choice is used only to remove nonconvex optimization pathologies; the phenomenon we isolate is *upper-curvature amplification along the trajectory.* Accordingly, our results should be read as a mechanism-level analysis within a stabilized optimization view of TD, rather than as a full convergence theory for stochastic nonconvex deep RL.

Within this setting, we retain the force-dominance stability lens while removing the separate global smoothness requirement used to certify inner descent. Concretely, we replace global $L$-smoothness with a generalized smoothness (curvature-growth) model in which the Hessian magnitude is controlled by a nondecreasing pro-

file $\ell(\cdot)$ evaluated at the *local gradient scale* (Zhang et al., 2020; Tyurin, 2024). We quantify target drift through a cross-sensitivity (partial-gradient drift) bound on how $\nabla_w H(\theta, \cdot)$ changes with $\theta$, which enters only through the forcing term in the outer recursion. Technically, we extend generalized-smoothness descent certificates from static objectives to the coupled TD recursion and derive a curvature-checked stepsize rule that guarantees inner descent without global curvature constants. This allows the outer force-dominance contraction argument of (Asadi et al., 2023b) to go through unchanged, while making the admissible constant stepsize and the resulting linear rate explicitly depend on the worst gradient scale encountered along the run. In particular, the theory yields a sharp stability scaling law: as the trajectory gradient envelope grows, the largest safe constant stepsize shrinks according to $\ell(\cdot)$, and maintaining a fixed contraction requires the inner effort $K$ to scale proportionally to the induced curvature amplification.

## 1.1 Contributions

We strengthen the optimization-theoretic convergence theory of temporal-difference learning with target networks by removing the global smoothness requirement on the inner objective and replacing it with a curvature-growth model that permits unbounded, trajectory-dependent upper curvature. Concretely, our contributions are:

- **Force-dominance beyond global smoothness.** Extending the force-dominance stability lens of Asadi et al. (2023b), we prove global linear convergence for inexact TD with $K$ inner gradient steps per target refresh under generalized smoothness.

- **Constant stepsizes via curvature checks.** We analyze a curvature-checked stepsize rule for the inner updates and show that a constant baseline stepsize achieves global stability and linear convergence under generalized smoothness, provided it satisfies a single trajectory-dependent admissibility condition governed by the maximum gradient magnitude along the run.

- **Trajectory-dependent stability scaling.** We identify the trajectory gradient envelope as the key stability quantity under curvature growth, yielding an explicit scaling law: the largest admissible constant stepsize decays with curvature amplification at the worst gradient scale, and maintaining a fixed contraction requires $K$ to scale proportionally with this amplification.

- **Classical smooth rates as a special case.** For uniformly smooth objectives (constant curvature profile) we recover the condition-number rates of (Asadi et al., 2023b); under curvature growth we quantify the stability penalty from high-curvature initialization and large early Bellman residual–induced gradient magnitudes in strongly convex, stabilized inner problems.

## 2 Problem Setting

Reinforcement learning (RL) studies artificial agents that learn through trial and error (Sutton et al., 1998). In this paper, we focus on the *prediction* setting, where the agent is interested in estimating the long-term goodness (value) of its states. This setting is mathematically formalized by a Markov reward process (MRP) (Puterman, 2014). We consider the discounted infinite-horizon case, specified by the tuple $\langle \mathcal{S}, R, P, \gamma \rangle$, where $\mathcal{S}$ is the set of states, $R : \mathcal{S} \to \mathbb{R}$ denotes the reward when transitioning out of a state, $P : \mathcal{S} \to \Delta(\mathcal{S})$ is the transition kernel (denoted by $P(s' \mid s)$), and $\gamma \in (0, 1)$ is the discount factor.

The goal is to approximate the state-value function, $v(s) \triangleq \mathbb{E}[\sum_{t=0}^{\infty} \gamma^t r_t \mid s_0 = s]$. Define the Bellman operator $T$ by $(Tv)(s) \triangleq R(s) + \sum_{s' \in \mathcal{S}} \gamma P(s' \mid s) v(s')$, equivalently $Tv = R + \gamma Pv$. In large-scale RL problems, $|\mathcal{S}|$ can be enormous, making tabular approaches infeasible. We therefore focus on parameterized function approximation and seek parameters $\theta$ such that the learned value function $v(\cdot; \theta)$ approximates $v(\cdot)$ well.

A fundamental and widely used approach is temporal-difference (TD) learning (Sutton, 1988). Given a sample $\langle s, r, s' \rangle$ with $s' \sim P(\cdot \mid s)$, TD learning algorithm updates the parameters of the approximate value function as follows:

$$\theta_{t+1} \leftarrow \theta_t + \alpha \left( r + \gamma v(s'; \theta_t) - v(s; \theta_t) \right) \nabla_\theta v(s; \theta_t), \tag{2}$$

where $\alpha > 0$ is a stepsize and $\nabla_\theta v(s; \theta_t)$ denotes the partial gradient of $v(s; \theta)$ with respect to $\theta$. The bootstrapped target $r + \gamma v(s'; \theta_t)$ is a one-step lookahead estimate and can be viewed as a sample of the right-hand side of the Bellman equation. TD is central not only as a standalone prediction algorithm but also because many modern RL methods inherit this bootstrapping structure, often combined with function approximation and target networks.

To place stability questions in context, let $d(\cdot)$ denote the stationary distribution of the Markov chain, defined by $\forall s' \in \mathcal{S}, \quad \sum_{s \in \mathcal{S}} d(s) P(s' \mid s) = d(s')$. Classical results show that, under linear function approximation, TD can be convergent under appropriate sampling conditions, notably when states are sampled from $d(\cdot)$ (Tsitsiklis & Van Roy, 1996). At the same time, TD can exhibit divergence when states are sampled from an arbitrary distribution rather than the stationary distribution of the Markov chain. First identified by Tsitsiklis & Van Roy (1996), this example is a very simple Markov chain with two non-terminal states and zero rewards. In this paper, we focus on the Markov reward process (MRP) setting, which could be thought of as a Markov decision process (MDP) with a single action. TD can display divergence even in the MRP setting (Tsitsiklis & Van Roy, 1996; Sutton et al., 1998), which indicates that the presence of multiple actions is not the core reason behind TD misbehavior (Sutton et al., 1998) – Divergence can manifest itself even in the more specific case of single action.

Moreover, little is known about the convergence guarantees of TD under alternative function approximators, or even when the TD update rule is slightly modified. These observations motivate studying TD in a way that explicitly exposes the algorithmic structure used in practice, rather than restricting attention to special linear regimes.

## 2.1 TD Learning as Two-level Optimization

A recent optimization-perspective analysis Asadi et al. (2023b) recasts practical TD-style procedures as a two-level scheme with two parameter roles: a *target* parameter $\theta_t$ is held fixed while an *online* parameter $w$ is updated (to approximately minimize a target-induced objective), after which the target is refreshed by setting $\theta_{t+1} \approx w$. More precisely, following Asadi et al. (2023b), we start from the single-parameter TD update in (2) and adopt the standard target-network decoupling (Mnih et al., 2015) that separates target parameters $\theta$ from online optimization parameters $w$ and updates $\theta$ less frequently. In particular, at outer iteration $t$ we freeze $\theta_t$ and perform $K$ gradient steps on $w$ while keeping $\theta_t$ fixed:

$$w^{t,k+1} \leftarrow w^{t,k} + \alpha \Big( r + \gamma\, v(s'; \theta^t) - v(s; w^{t,k}) \Big) \nabla_w v(s; w^{t,k}) \tag{3}$$

and then updates the target parameter $\theta^{t+1} \leftarrow w^{t,K}$ before moving to the next iteration $t+1$. Here $K$ is a hyper-parameter, where $K = 1$ takes us back to the original TD update (2). Observe that the dependence of $v(s'; \theta^t)$ to the optimization parameter $w$ is ignored in this update, despite the fact that an implicit dependence is present due to the final step $\theta^{t+1} \leftarrow w^{t,K}$. This means that the objective function being optimized is made up of two separate input variables. We now define:

$$H(\theta, w) \triangleq \frac{1}{2} \sum_s d(s) \Big( \mathbb{E}_{r,s'}\big[ r + \gamma v(s'; \theta) \big] - v(s; w) \Big)^2, \tag{4}$$

where $d(.)$ is allowed to be an arbitrary distribution, not just the stationary-state distribution of the Markov chain. Moreover, the scalar term $r + \gamma\, v(s', \theta^t) - v(s; w^{t,k})$ in (3) is the one-step TD error (Bellman residual), so $\nabla_w H(\theta, w)$ aggregates Bellman residuals weighted by the critic Jacobian $\nabla_w v(s; w)$; accordingly, $\|\nabla_w H(\theta, w)\|$ should be read as a Jacobian-weighted residual scale. Hence, TD, DQN, and similar algorithms could best be thought of as learning algorithms that proceed by approximately solv-

Observe that the update direction in (3) is an unbiased stochastic estimate of the negative partial gradient $-\nabla_w H(\theta, w)$.

---

**Algorithm 1:** Inexact TD via $K$ Inner Gradient Steps

**Input:** $\theta^0$, $T$, $K$, $\{\alpha_{t,k}\}$
**for** $t = 0$ **to** $T - 1$ **do**
    $w^{t,0} \leftarrow \theta^t$;
    **for** $k = 0$ **to** $K - 1$ **do**
        $w^{t,k+1} \leftarrow w^{t,k} - \alpha_{t,k} \nabla_w H(\theta^t, w^{t,k})$;
    $\theta^{t+1} \leftarrow w^{t,K}$;
**return** $\theta^T$;

ing for the following sequence of optimization problems, $\theta_{t+1} \approx \arg\min_w H(\theta_t, w)$, using first-order optimization techniques. This optimization perspective is useful conceptually because it accentuates the unusual property of this iterative process, namely that the first argument of the objective $H$ hinges on the output of the previous iteration. It also helps motivate optimizer design questions in RL (Asadi et al., 2023a). Moreover, the general form of this optimization process allows for using alternative forms of loss functions such as the Huber loss (Mnih et al., 2015), the logistic loss (Bas-Serrano et al., 2021), or the entropy (Precup & Sutton, 1997), as well as various forms of function approximation such as linear functions or deep neural networks. Each combination of loss functions and function approximators yields a different $H$, but one that is always comprised of a function $H$ with two inputs.

In practice, the inner problem induced by a frozen target parameter $\theta^t$ is not solved exactly. Instead, one performs a finite number $K \geq 1$ of first-order updates on an online parameter $w$ to approximately minimize the target-induced objective $H(\theta^t, \cdot)$, and then refreshes the target by setting $\theta^{t+1} \leftarrow w^{t,K}$. Concretely, the inexact recursion is given in Algorithm (1).

The theoretical difficulty is that the inner problem is only solved approximately (finite $K$), so one must control how much descent the inner updates achieve under the geometry of $H(\theta^t, \cdot)$ and then propagate this control through the outer recursion $\theta^{t+1} \leftarrow w^{t,K}$.

## 3 Generalized-Smoothness Preliminaries

We analyze inexact TD through the inner objectives

$$f_t(w) \triangleq H(\theta_t, w), \qquad w_{t,0} \triangleq \theta_t,$$

where $t$ indexes the outer (target) iterate. This section collects the generalized-smoothness facts we use to certify first-order descent for inner gradient steps without assuming a global $L$-smoothness constant. The two takeaways are: (i) gradient differences can be controlled by a curvature-growth profile evaluated at the local gradient scale, and (ii) the classical $L$-smooth descent lemma is replaced by an integral bound whose magnitude depends on that same scale.

**Assumption 3.1** (Generalized smoothness / curvature growth (Tyurin, 2024)). *Fix $t$. The function $f_t : \mathbb{R}^d \to \mathbb{R} \cup \{+\infty\}$ is $\ell$-smooth on $\mathcal{W}$ if $f_t$ is twice differentiable on $\mathcal{W}$, continuous on the closure of $\mathcal{W}$, and there exists a non-decreasing, positive, locally Lipschitz function $\ell : [0, \infty) \to (0, \infty)$ such that*

$$\left\|\nabla^2 f_t(w)\right\|_{\mathrm{op}} \leq \ell\big(\|\nabla f_t(w)\|\big) \qquad \text{for all } w \in \mathcal{W}. \tag{5}$$

Assumption 3.1 strictly generalizes classical $L$-smoothness (take $\ell \equiv L$) and polynomial curvature-growth models of the form $\|\nabla^2 f_t(w)\|_{\mathrm{op}} \leq L_0 + L_1 \|\nabla f_t(w)\|^p$. Although (5) is stated using Hessians, our analysis uses it only to derive gradient-based inequalities; in particular, we will later invoke a purely first-order consequence (Lemma 3.4) that can be taken as an alternative assumption when $f_t$ is not twice differentiable.

**Assumption 3.2** (Lower boundedness). *For each fixed $t$, there exists $f_t^\star \in \mathbb{R}$ such that $f_t(w) \geq f_t^\star$ for all $w \in \mathcal{W}$.*

Following Tyurin (2024), we introduce a modulus that converts the curvature-growth profile $\ell(\cdot)$ into explicit bounds on gradient variation.

**Definition 3.3** ($q$–function). *Assume Assumption 3.1. For $a \geq 0$, define*

$$q(s; a) = \int_0^s \frac{dv}{\ell(a + v)}, \qquad q_{\max}(a) := \int_0^\infty \frac{dv}{\ell(a + v)} \in (0, \infty]. \tag{6}$$

**Proposition 1** (Tyurin (2024)). *The function $q(\cdot; a)$ is invertible, differentiable, positive, and strongly increasing, and its inverse $q^{-1}(\cdot; a) : [0, q_{\max}(a)) \to \mathbb{R}_+$ is also differentiable, positive, and strongly increasing.*

The inverse $q^{-1}$ appears as an intermediate modulus in the generalized-smoothness bounds below. Importantly, our final stability and stepsize conditions will be expressed directly through $\ell(\cdot)$ evaluated at trajectory-dependent gradient scales.

**Lemma 3.4.** *Fix $t$. For all $w, u \in \mathcal{W}$ with $\|u-w\| \in [0, q_{\max}(\|\nabla f_t(w)\|))$, if $f_t$ is $\ell$–smooth (Assumption 3.1), then*

$$\left\| \nabla f_t(u) - \nabla f_t(w) \right\| \leq q^{-1}\big(\|u-w\| \, ; \|\nabla f_t(w)\|\big), \tag{7}$$

*where $q$ and $q_{\max}(\cdot)$ are defined in Definition 3.3.*

**Remark.** When $f_t$ is not twice differentiable, one may replace Assumption 3.1 by directly assuming (7); the subsequent analysis remains valid. We keep Assumption 3.1 because it provides an interpretable curvature-growth model.

For $L$-smooth objectives, one has $f_t(u) \leq f_t(w) + \langle \nabla f_t(w), u-w \rangle + \frac{L}{2}\|u-w\|^2$. Under generalized smoothness, the quadratic term is replaced by an integral modulus:

**Lemma 3.5** (Tyurin (2024)). *Fix $t$. For all $w, u \in \mathcal{W}$ with $\|u-w\| \in [0, q_{\max}(\|\nabla f_t(w)\|))$, if $f_t$ is $\ell$–smooth (Assumption 3.1), then*

$$f_t(u) \leq f_t(w) + \langle \nabla f_t(w), \, u - w \rangle + \int_0^{\|u-w\|} q^{-1}\big(\tau; \|\nabla f_t(w)\|\big) \, d\tau. \tag{8}$$

Lemmas 3.4–3.5 are the only generalized-smoothness tools we require later. They play the role of the classical smoothness inequalities, but crucially their strength depends on the local gradient scale through $\ell(\cdot)$, which is what ultimately drives the trajectory-dependent stepsize and rate behavior in our TD analysis.

# 4 Convergence of TD with Generalized Smoothness

*All proofs are deferred to the Appendix.*

Our goal is to establish TD convergence guarantees without requiring a uniform bound on the Hessian of the inner objective. We work in the standard two-level optimization view of TD with target networks, and we isolate the failure mode that is orthogonal to nonconvexity: *heterogeneous upper curvature along the training trajectory.* Accordingly, we retain the stabilized regime in which the inner problem is well-posed (strongly convex in the online variable), and we remove the separate global $L$-smoothness assumption used in prior force-dominance analyses to certify inner descent.

We study Algorithm 1 for a general function $H : \mathbb{R}^d \times \mathbb{R}^d \to \mathbb{R}$ under the following assumptions.

**Assumption 4.1** ($F_\theta$-Lipschitz partial gradient in $\theta$). *There exists a constant $F_\theta \geq 0$ such that for all $\theta_1, \theta_2$ and all $w$,*

$$\left\| \nabla_w H(\theta_1, w) - \nabla_w H(\theta_2, w) \right\| \leq F_\theta \|\theta_1 - \theta_2\|.$$

**Assumption 4.2** ($F_w$-strong convexity in $w$). *For every fixed $\theta$, the function $w \mapsto H(\theta, w)$ is $F_w$-strongly convex. Equivalently, there exists a constant $F_w > 0$ such that for all $w_1, w_2$ and all $\theta$,*

$$\big(\nabla_w H(\theta, w_1) - \nabla_w H(\theta, w_2)\big)^\top (w_1 - w_2) \geq F_w \|w_1 - w_2\|^2.$$

**Assumption 4.3** (Fixed point). *There exists $\theta^\star \in \mathbb{R}^d$ such that $\nabla_w H(\theta^\star, \theta^\star) = 0$.*

We analyze TD under generalized smoothness (Assumption 3.1) in three stages. First, we construct a *curvature-checked* stepsize that guarantees descent without global $L$-smoothness. Second, we combine this descent certificate with strong convexity in $w$ to obtain an inner contraction with a forcing term. Finally, we couple the inner and outer dynamics and obtain *global linear convergence* under the *force-dominance* condition $F_\theta < F_w$, with a constant baseline stepsize calibrated to the trajectory-dependent curvature growth.

### 4.1 The Curvature-Checked Stepsize

Standard analyses rely on global $L$-smoothness, which yields the quadratic upper bound $f_t(u) \leq f_t(w) + \langle \nabla f_t(w), u - w \rangle + \frac{L}{2}\|u - w\|^2$, $\forall w, u \in \mathcal{W}$. This inequality justifies a constant stepsize $\gamma \leq 1/L$: for the update $u = w - \gamma \nabla f_t(w)$, the linear decrease $-\gamma \|\nabla f_t(w)\|^2$ dominates the quadratic curvature penalty $\frac{L}{2}\gamma^2\|\nabla f_t(w)\|^2$. Under generalized smoothness (Assumption 3.1), the effective curvature is not constant; it is governed by the curvature-growth profile $\ell(\|\nabla f_t(w)\|)$ and can become large in steep regions. Consequently, a naively fixed stepsize need not guarantee descent.

We therefore adopt the clipped constant-step rule of the form $\gamma_{t,k} = \min\{\gamma_0, \gamma_{t,k}^{\text{safe}}\}$, which separates optimization speed from geometric safety. The baseline $\gamma_0$ is the step we would like to use throughout the run, since it yields a uniform inner contraction once descent is certified. The cap $\gamma_{t,k}^{\text{safe}}$ is computed from the generalized-smoothness model to guarantee a net decrease for $w_{t,k+1} = w_{t,k} - \gamma_{t,k}\nabla f_t(w_{t,k})$, even when $\|\nabla^2 f_t\|$ is unbounded. Later, we show that along stable trajectories the gradients admit a uniform envelope $M$, which implies a uniform positive lower bound on $\gamma_{t,k}^{\text{safe}}$; choosing $\gamma_0$ below this bound makes clipping inactive, and the method reduces to constant-stepsize gradient descent. Then we derive a uniform linear contraction.

**Strategy.** Assumption 3.1 provides a generalized upper model in which the remainder term is controlled through the inverse profile $q^{-1}(\cdot;\cdot)$ (Lemma 3.5). We choose $\gamma_{t,k}$ so that this remainder consumes at most a fixed fraction of the first-order decrease. To bound the integral remainder in (8), we use the monotonicity of $q^{-1}(\tau;a)$ in $\tau$, yielding the rectangular bound $\int_0^r q^{-1}(\tau;a)\,d\tau \leq r\,q^{-1}(r;a)$, $r \in [0, q_{\max}(a))$.

**Definition 4.4** (Constant target schedule). *For a fixed $\gamma_0 > 0$, we set $\overline{\gamma}_n \equiv \gamma_0$ for all global steps $n = tK + k$.*

**Curvature-checked stepsize.** Let $g_{t,k} \triangleq \|\nabla f_t(w_{t,k})\|$. When $g_{t,k} = 0$, any stepsize yields no motion, so we use the baseline schedule. When $g_{t,k} > 0$, we define a *safe cap* from the curvature-growth profile so that the generalized remainder term is controlled relative to the linear decrease.

**Definition 4.5** (Curvature-checked constant stepsize). *At inner step $(t, k)$, let $g_{t,k} \triangleq \|\nabla f_t(w_{t,k})\|$ and $n = tK + k$. If $g_{t,k} = 0$, set $\gamma_{t,k} = \overline{\gamma}_n = \gamma_0$. If $g_{t,k} > 0$, define*

$$\gamma_{t,k}^{\text{safe}} \triangleq \frac{1}{g_{t,k}} q\left(\frac{g_{t,k}}{2}; g_{t,k}\right) = \frac{1}{g_{t,k}} \int_0^{g_{t,k}/2} \frac{d\tau}{\ell(g_{t,k} + \tau)} = \int_0^{1/2} \frac{dv}{\ell\big((1+v)g_{t,k}\big)}, \tag{9}$$

*and set the actual stepsize to be*

$$\gamma_{t,k} \triangleq \min\{\overline{\gamma}_n, \gamma_{t,k}^{\text{safe}}\} = \min\{\gamma_0, \gamma_{t,k}^{\text{safe}}\}. \tag{10}$$

**Remark 1.** Define the remainder ratio $c_{t,k} \triangleq \frac{q^{-1}\big(\gamma_{t,k} g_{t,k}; g_{t,k}\big)}{g_{t,k}}$. In the classical $L$-smooth case, one has $q^{-1}(r;g) = Lr$, so the remainder ratio reduces to $c_{t,k} = L\gamma_{t,k}$. Requiring $c_{t,k} \leq \frac{1}{2}$ therefore yields a familiar constant-stepsize constraint of order $1/L$ (e.g., $\gamma_{t,k} \lesssim 1/L$), which is sufficient to guarantee a uniform fraction of descent. In our setting, $q^{-1}$ is nonlinear and depends on the gradient scale; the choice (9) is constructed so that the generalized curvature remainder in (8) is controlled relative to the linear term, ensuring that it consumes at most a fixed fraction of the predicted first-order decrease.

**Lemma 4.6** (One-step descent certificate). *Under Assumption 3.1 and choosing $\gamma_{t,k}$ by Definition 4.5, each inner update satisfies*

$$f_t(w_{t,k+1}) - f_t(w_{t,k}) \leq -\frac{\gamma_{t,k}}{2} \|\nabla f_t(w_{t,k})\|^2 = -\frac{1}{2\gamma_{t,k}} \|w_{t,k+1} - w_{t,k}\|^2. \tag{11}$$

**Remark 2.** Lemma 4.6 replaces the smoothness-based *choose $\gamma \leq 1/L$* step used in standard analyses: it certifies descent without global Lipschitz gradients in $w$. In the next section, we combine this certificate with strong convexity to obtain an inner contraction, and then lift it to an outer force-dominance contraction for the inexact $K$-step TD scheme.

## 4.2 The Inner Contraction

Section 4.1 established a local descent certificate for each inner update: the inner loop decreases $f_t(w) = H(\theta_t, w)$ even when $f_t$ does not admit a global smoothness constant in $w$. In this section we show that the inner iterates contract toward the TD fixed point $\theta^\star$ up to an additive forcing term. The key additional ingredient is strong convexity in $w$ (Assumption 4.2), which converts descent into geometric contraction.

A technical subtlety is that, at outer time $t$, the minimizer of $w \mapsto H(\theta_t, w)$ need not coincide with the fixed point $\theta^\star$. We measure this mismatch at the reference point $\theta^\star$ via $\Delta_t \triangleq \|\nabla_w H(\theta_t, \theta^\star)\|$. By Assumption 4.3, $\nabla_w H(\theta^\star, \theta^\star) = 0$, so $\Delta_t = \|\nabla_w H(\theta_t, \theta^\star) - \nabla_w H(\theta^\star, \theta^\star)\|$.

**Proposition 2** (Key suboptimality bound). *Under assumptions 4.2 and 4.3 and choosing $\gamma_{t,k}$ by Definition 4.5, for every inner step $(t, k)$,*

$$H(\theta_t, \theta^\star) - H(\theta_t, w_{t,k}) \;\leq\; \frac{\Delta_t^2}{2F_w} - \frac{1}{2\gamma_{t,k}}\big\|w_{t,k+1} - w_{t,k}\big\|^2. \tag{12}$$

Proposition 2 is where the descent certificate interacts with strong convexity. It bounds the inner-loop suboptimality at $w_{t,k}$ in terms of (i) the reference mismatch $\Delta_t$ and (ii) the step displacement $\|w_{t,k+1} - w_{t,k}\|$. The key consequence is that when we expand the squared distance to $\theta^\star$ after one gradient step, the potentially harmful cross term is controlled by this suboptimality bound while the step-length term from Lemma 4.6 cancels, yielding a clean one-step contraction up to an additive forcing term.

**Proposition 3** (Inner-step contraction with forcing). *Define the effective contraction rate $\kappa_{t,k} \triangleq F_w \gamma_{t,k}$. Then, for all $t$ and all $k$,*

$$\|w_{t,k+1} - \theta^\star\|^2 \;\leq\; (1 - \kappa_{t,k})\|w_{t,k} - \theta^\star\|^2 + \kappa_{t,k}\frac{\Delta_t^2}{F_w^2}. \tag{13}$$

**Remark 3.** The additive term $\kappa_{t,k}\Delta_t^2/F_w^2$ appears because $\theta^\star$ is a fixed point of the two-level scheme, not necessarily the minimizer of $w \mapsto H(\theta_t, w)$ for general $\theta_t$. When $\Delta_t = 0$, the inner loop contracts directly toward $\theta^\star$.

Proposition 3 controls a single inner gradient step. The TD update runs $K$ such steps with the target frozen, and then refreshes the target via $\theta_{t+1} \leftarrow w_{t,K}$. Iterating the one-step contraction yields an outer recursion.

**Proposition 4** (Outer recursion). *Define the cumulative decay*

$$\chi_{t,K} \triangleq \prod_{k=0}^{K-1}(1 - \kappa_{t,k}), \qquad \eta_t^2 \triangleq \frac{\Delta_t^2}{F_w^2}.$$

*Then,*

$$\|\theta_{t+1} - \theta^\star\|^2 \;\leq\; \chi_{t,K}\|\theta_t - \theta^\star\|^2 + \big(1 - \chi_{t,K}\big)\eta_t^2. \tag{14}$$

The recursion in (14) is the central interface between inner optimization and outer stability: $\chi_{t,K}$ captures the cumulative inner contraction over $K$ steps, while $\eta_t$ captures the mismatch-driven forcing. We next bound $\eta_t$ using Assumption 4.1 and obtain a uniform contraction under force dominance.

## 4.3 From the Outer Recursion to Fixed-Point Convergence

Proposition 4 yields the fundamental inexact recursion (14). To turn it into a uniform linear rate, we need two ingredients: (i) force dominance $F_\theta < F_w$, which controls the forcing term $\eta_t$, and (ii) a uniform inner contraction factor, which we obtain once clipping is inactive and $\gamma_{t,k} \equiv \gamma_0$.

Using Assumption 4.1 with $w = \theta^\star$ and $\theta_2 = \theta^\star$ yields $\Delta_t = \|\nabla_w H(\theta_t, \theta^\star)\| \leq F_\theta\|\theta_t - \theta^\star\|$. Since $\eta_t^2 = \Delta_t^2/F_w^2$, defining $\eta \triangleq F_\theta/F_w$ gives $\eta_t^2 \leq \eta^2\|\theta_t - \theta^\star\|^2$. Substituting into (14) yields

$$\|\theta_{t+1} - \theta^\star\|^2 \leq \Big(\eta^2 + (1 - \eta^2)\chi_{t,K}\Big)\|\theta_t - \theta^\star\|^2. \tag{15}$$

Under force dominance $F_\theta < F_w$, we have $\eta \in (0, 1)$.

It remains to ensure a uniform lower bound on the safety cap. Once $\gamma_{t,k}^{\mathrm{safe}} \geq \gamma_{\min}^{\mathrm{safe}} > 0$ holds for all iterates, choosing $\gamma_0 \leq \gamma_{\min}^{\mathrm{safe}}$ guarantees $\gamma_{t,k} = \gamma_0$ everywhere and $\chi_{t,K} = (1 - F_w\gamma_0)^K$. Since $\gamma_{t,k}^{\mathrm{safe}}$ depends on the local gradient scale, we obtain this by first showing that the iterates remain in a bounded region, which implies a uniform bound on the gradient norms along the trajectory.

**Lemma 4.7** (Bounded iterates and a uniform gradient envelope). *Assume $H \in \mathcal{C}^1(\mathbb{R}^d \times \mathbb{R}^d)$, Propositions 2– 4 holds, the force-dominance condition $F_\theta < F_w$ holds, and the constant baseline stepsize satisfies $\gamma_0 < 1/F_w$. Let $R \triangleq \|\theta_0 - \theta^\star\|$ and define*

$$M \triangleq \sup\Big\{\|\nabla_w H(\theta, w)\| : \|\theta - \theta^\star\| \leq R, \ \|w - \theta^\star\| \leq R\Big\}. \tag{16}$$

*Then the following hold: (i) (**Bounded outer iterates**) For all $t \geq 0$, $\|\theta_t - \theta^\star\| \leq R$, (ii) (**Bounded inner iterates**) For all $t \geq 0$ and all $k \in \{0, \ldots, K\}$, $\|w_{t,k} - \theta^\star\| \leq R$, (iii) (**Uniform gradient envelope**) The constant $M$ in (16) is finite and for all $t, k$, i.e, $g_{t,k} = \|\nabla f_t(w_{t,k})\| = \|\nabla_w H(\theta_t, w_{t,k})\| \leq M < \infty$.*

# 5 A Unifying Theory of Convergence Under $\ell$-Smoothness

*All proofs are deferred to the Appendix.*

This section packages our constant-stepsize theory into one theorem and then specializes it across canonical curvature-growth regimes. The punchline is simple: *generalized smoothness does not preclude linear convergence*. It changes the admissibility law for a constant stepsize from a *global* curvature constant to a *trajectory-dependent* curvature amplification captured by a gradient envelope. This yields an explicit scaling law for the required inner effort $K$.

**Setup.** We study the inexact TD recursion (Algorithm 1) with $K \geq 1$ inner steps and the curvature-checked constant stepsize

$$\gamma_{t,k} = \min\{\gamma_0, \gamma_{t,k}^{\mathrm{safe}}\}, \qquad \gamma_{t,k}^{\mathrm{safe}} = \frac{1}{g_{t,k}} \int_0^{g_{t,k}/2} \frac{d\tau}{\ell(g_{t,k} + \tau)}, \qquad g_{t,k} = \|\nabla f_t(w_{t,k})\|.$$

The curvature check guarantees one-step descent unconditionally. The question in this section is when the check becomes *inactive*, so that the method effectively runs with a *single constant* stepsize and admits a *uniform linear rate*.

**A uniform admissibility threshold from the gradient envelope.** With a uniform gradient envelope in hand, we can lower bound $\gamma_{t,k}^{\mathrm{safe}}$ uniformly over all iterates. By Definition 4.5, for $g_{t,k} > 0$ we have $\gamma_{t,k}^{\mathrm{safe}} = \frac{1}{g_{t,k}} \int_0^{g_{t,k}/2} \frac{d\tau}{\ell(g_{t,k} + \tau)}$. Using $g_{t,k} \leq M$ from Lemma 4.7 and monotonicity of $\ell(\cdot)$, for all $\tau \in [0, g_{t,k}/2]$ we have $g_{t,k} + \tau \leq \frac{3}{2} g_{t,k} \leq \frac{3}{2} M$, hence $\ell(g_{t,k} + \tau) \leq \ell(3M/2)$ and therefore

$$\gamma_{t,k}^{\mathrm{safe}} \geq \frac{1}{g_{t,k}} \int_0^{g_{t,k}/2} \frac{d\tau}{\ell(3M/2)} = \frac{1}{2\,\ell(3M/2)} \triangleq \gamma_{\min}^{\mathrm{safe}}. \tag{17}$$

Consequently, choosing $\gamma_0 \leq \gamma_{\min}^{\mathrm{safe}}$[2] implies $\gamma_{t,k} = \gamma_0$ for all $t, k$, i.e., the curvature check never clips and the inner loop runs with a constant step. Hence $\kappa_{t,k} = F_w\gamma_0 \triangleq \kappa \in (0, 1)$. Therefore

$$\chi_{t,K} = \prod_{k=0}^{K-1} (1 - \kappa) = (1 - \kappa)^K \triangleq \chi_K \in (0, 1), \tag{18}$$

is constant in $t$. Plugging $\chi_{t,K} \equiv \chi_K$ into (15) yields a uniform contraction. We now state the unifying theorem.

---

[2]In the $\mathcal{C}^2$ regime considered throughout our generalized-smoothness analysis, Assumption 4.2 (strong convexity in first-order form) guarantees that for any fixed target, the objective satisfies $\nabla_w^2 H(\theta, \cdot) \succeq F_w I$. Recalling that $f_t(w) \triangleq H(\theta_t, w)$, Assumption 3.1 upper bounds the Hessian norm $\|\nabla_w^2 H(\theta, w)\|_{\mathrm{op}}$ by the profile $\ell(\|\nabla_w H(\theta, w)\|)$. Since the maximum eigenvalue must bound the minimum eigenvalue, this forces $\ell(\|\nabla_w H(\theta, w)\|) \geq F_w$ along the trajectory. Consequently, because $\ell$ is non-decreasing, the admissibility condition $\gamma_0 \leq [2\,\ell(3M/2)]^{-1}$ already strictly enforces $\gamma_0 \leq 1/(2F_w) < 1/F_w$.

**Theorem 5.1** (Unifying constant-stepsize convergence under $\ell$-smoothness)**.** *Assume Assumptions 3.1, 4.1, 4.2, 4.3, and suppose $H \in \mathcal{C}^1(\mathbb{R}^d \times \mathbb{R}^d)$. Let $M$ be the gradient envelope from Lemma 4.7. Assume force dominance $F_\theta < F_w$ and define $\eta \triangleq F_\theta/F_w \in (0,1)$. Choose a constant baseline $\gamma_0 > 0$ satisfying*

$$\gamma_0 \leq \frac{1}{2\,\ell(3M/2)}. \tag{19}$$

*Run Algorithm 1 with stepsizes $\gamma_{t,k} = \min\{\gamma_0, \gamma_{t,k}^{\mathrm{safe}}\}$. Then the curvature check never clips, $\gamma_{t,k} \equiv \gamma_0$, and the outer iterates contract linearly:*

$$\|\theta_{t+1} - \theta^\star\|^2 \leq \rho\,\|\theta_t - \theta^\star\|^2, \qquad \rho \triangleq \eta^2 + (1-\eta^2)(1-F_w\gamma_0)^K \in (0,1). \tag{20}$$

*Consequently, $\|\theta_t - \theta^\star\|^2 \leq \rho^{\,t}\|\theta_0 - \theta^\star\|^2$.*

---

Theorem 5.1 establishes three key trajectory-dependent properties governed by the envelope $M$:

1. **Admissible constant step.** The largest globally admissible constant stepsize scales as

$$\gamma_0^{\mathrm{max}} \;=\; \frac{1}{2\,\ell(3M/2)}. \tag{21}$$

2. **Trajectory-dependent condition number.** Generalized smoothness replaces the classical $L/F_w$ (Asadi et al., 2023b) by the trajectory-dependent amplification

$$\mathcal{B}(M) \;\triangleq\; \frac{\ell(3M/2)}{F_w}. \tag{22}$$

3. **The stability scaling law (inner-effort threshold).** If we saturate $\gamma_0$ at its curvature-growth limit $\gamma_0 = [2\ell(3M/2)]^{-1}$, then

$$(1 - F_w\gamma_0)^K = \Big(1 - \frac{1}{2\mathcal{B}(M)}\Big)^K \;\leq\; \exp\!\Big(-\frac{K}{2\mathcal{B}(M)}\Big). \tag{23}$$

Thus, achieving a fixed-strength inner contraction requires

$$K \;\gtrsim\; \mathcal{B}(M) \;=\; \frac{\ell(3M/2)}{F_w}. \tag{24}$$

---

Because $M$ strictly bounds the gradient envelope (Lemma 4.7), and this gradient represents the expected TD update direction (a Jacobian-weighted Bellman residual), $M$ essentially captures the worst-case geometry-weighted Bellman residual scale encountered during training. These relations make the source of initialization sensitivity explicit: large early TD errors (the residual term in (3)) inflate the trajectory gradient envelope $M$, which increases $\ell(3M/2)$, shrinks $\gamma_0^{\mathrm{max}}$, and raises the inner-effort threshold (24).

**Remark 4.** The constraint $\gamma_0 \leq [2\ell(3M/2)]^{-1}$ is the *uniform linear-rate admissibility law*: it guarantees that one constant baseline step is valid everywhere along the trajectory, summarized by the gradient envelope $M$. This is exactly where generalized smoothness departs from global $L$-smoothness: the admissible constant step is controlled by the worst gradient scale reached during learning, rather than curvature at the optimum. When $\gamma_0$ exceeds the local stability threshold, a global descent argument for constant-step gradient methods can fail and instability can occur, closely related to edge-of-stability behavior observed in large-scale training (Cohen et al., 2021). In contrast, Definition 4.5 is *fail-safe*: if $\gamma_0$ is too aggressive for (19), the update clips to $\gamma_{t,k}^{\mathrm{safe}}$ and preserves the one-step descent certificate. The tradeoff is that a *uniform* contraction factor need not hold, so one must analyze the resulting variable-stepsize dynamics.

To derive explicit convergence rates for distinct smoothness classes, we now specialize (20) by saturating the admissibility condition in (19), i.e., $\gamma_0 = \frac{1}{2\,\ell(3M/2)}$. This makes the dependence on curvature amplification and inner effort $K$ explicit.

## 5.1 Regime I: Global $L$-Smoothness ($\ell(s) \equiv L$)

When $\ell(s) \equiv L$, the admissibility condition in (19) becomes $\gamma_0 \leq 1/(2L)$ in our certificate, and Theorem 5.1 yields $\rho = \eta^2 + (1 - \eta^2)\left(1 - \frac{F_w}{2L}\right)^K \leq \eta^2 + (1 - \eta^2)\exp\left(-K\frac{F_w}{2L}\right)$.

**Insight.** In the globally smooth regime, the stability geometry collapses to the classical condition-number scale $L/F_w$. The effect of $K$ is transparent: each outer step composes a $K$-step inner contraction, and force dominance transfers that progress to the target parameter. In the special case $\ell \equiv L$, one may replace the monotone-integral remainder bound by the exact quadratic smoothness model, recovering the classical admissible step $\gamma_0 = 1/L$ and the contraction factor $\rho = \eta^2 + (1 - \eta^2)\left(1 - \frac{F_w}{L}\right)^K$, matching Proposition 5 of Asadi et al. (2023b).[3]

## 5.2 Regime II: Affine Growth with $\ell(s) = L_0 + L_1 s$ and Initialization-Dependent Rates

Here $\ell(3M/2) = L_0 + \frac{3}{2}L_1 M \Rightarrow \gamma_0 \leq \frac{1}{2L_0 + 3L_1 M}$. Thus, Theorem 5.1 yields $\rho = \eta^2 + (1 - \eta^2)\left(1 - \frac{F_w}{2L_0 + 3L_1 M}\right)^K$.

**Insight.** The rate is linear but explicitly trajectory-dependent: the single scalar $M$ sets the curvature amplification $\ell(3M/2)$ and therefore caps the globally admissible constant step. The critical threshold reads $K \gtrsim (L_0 + \frac{3}{2}L_1 M)/F_w$: if $M$ is large, a fixed $K$ yields a weak inner contraction and the outer progress is dominated by the forcing ratio $\eta$; increasing $K$ restores a strong linear rate.

## 5.3 Regime III: Polynomial Growth with $\ell(s) = L_0 + L_1 s^p$, $p \in [0, 2]$

In this regime, $\ell(3M/2) = L_0 + L_1\left(\frac{3M}{2}\right)^p \Rightarrow \gamma_0 \leq \frac{1}{2\left(L_0 + L_1(3M/2)^p\right)}$. Hence, $\rho = \eta^2 + (1 - \eta^2)\left(1 - \frac{F_w}{2(L_0 + L_1(3M/2)^p)}\right)^K$.

**Insight.** The admissible constant step deteriorates polynomially with the envelope: $\gamma_0^{\max} = \Theta(M^{-p})$ once $M$ is large. Equivalently, the critical inner-effort threshold is $K \gtrsim (L_0 + L_1(3M/2)^p)/F_w$. For $p \leq 2$, this yields a controlled sensitivity law: curvature amplification does not explode faster than quadratically in the gradient scale, so a constant-step linear regime remains meaningful as long as the trajectory stays within a moderate envelope.

## 5.4 Regime IV: Superquadratic Growth with $\ell(s) = L_0 + L_1 s^p$, $p > 2$

This specialization yields the same expression for $p$ with $p > 2$.

**Insight.** Superquadratic growth produces a sharp degradation in admissible constant steps: $\gamma_0^{\max} = \Theta(M^{-p})$ collapses rapidly, and the critical inner effort scales as $K \gtrsim M^p/F_w$ (up to constants). Theorem 5.1 still guarantees linear convergence, but it predicts a practical fragility: if $M$ is large and $K$ is fixed, the inner contraction becomes negligible and progress is limited by the forcing ratio $\eta$. This delineates when constant-step TD is *theoretically stable but practically conservative*, motivating staged strategies such as: (i) run curvature-check in a burn-in phase to reduce the envelope, then switch to a constant-step linear regime, or (ii) use adaptive steps that track local curvature growth more tightly than a worst-case bound at $M$.

## 5.5 Regime V: Exponential Growth with $\ell(s) = L_0 + L_1 s^2 e^s$

In this regime, $\ell(3M/2) = L_0 + L_1\left(\frac{3M}{2}\right)^2 e^{3M/2} = L_0 + \frac{9}{4}L_1 M^2 e^{3M/2}$, so, $\gamma_0 \leq \frac{1}{2L_0 + \frac{9}{2}L_1 M^2 e^{3M/2}}$, $\rho = \eta^2 + (1 - \eta^2)\left(1 - \frac{F_w}{2L_0 + \frac{9}{2}L_1 M^2 e^{3M/2}}\right)^K$.

---

[3]Our $1/(2L)$ admissibility arises from a deliberately robust remainder bound that remains valid under curvature growth; tightening it in the constant-$L$ case recovers the classical constant exactly.

**Insight.** Here the admissible constant step decays essentially like $e^{-3M/2}/M^2$, and the critical inner effort $K \gtrsim \ell(3M/2)/F_w$ becomes astronomically large once $M$ is moderate. This regime makes the role of curvature checks concrete: a globally safe constant step can be so small as to be useless, while the curvature-checked rule remains stable in steep regions and recovers a constant-step linear regime once the trajectory enters a smaller envelope.

## 6 Related Work

**Target networks, inexact inner solves, and force-dominance views of TD.** A recent line of work has moved beyond purely operator-based analyses by treating TD with target networks as a two-level optimization recursion. In particular, Asadi et al. (2023b) formalize TD with frozen targets as an iterative optimization procedure and show that stability can be characterized through a *force-dominance* condition balancing inner curvature against target drift; their linear convergence guarantees, however, certify inner descent using a global Lipschitz-smoothness assumption on the inner objective. Complementing this view, Fellows et al. (2023) analyze *partially fitted* TD methods and show that slower target updates can mitigate poor conditioning of the TD Jacobian, providing a principled explanation for the stabilizing effect of target networks. More recently, Weissmann et al. (2026) study target update frequencies in Q-learning and derive finite-time tradeoffs induced by periodic target fixing, motivating principled update schedules. Collectively, these works position TD as a coupled two-level dynamical system in which both inner optimization dynamics and target drift determine stability.

**Stabilization under the deadly triad.** Several works revisit stability mechanisms for TD-style updates under bootstrapping, function approximation, and off-policy sampling. Zhang et al. (2021) analyze target-network mechanisms in linear settings and provide convergent variants that incorporate projection and regularization, establishing convergence to well-defined regularized fixed points in regimes where standard methods can diverge. At the same time, Manek & Kolter (2022) show that regularization alone does not universally prevent divergence and can introduce new instability modes, underscoring that stabilization is structural rather than incidental. Gradient-TD variants provide alternative routes to soundness while remaining close to practical TD updates; for example, Ghiassian et al. (2020) propose TDRC, a regularized correction that improves stability in challenging off-policy regimes.

**Optimization under curvature growth and generalized smoothness.** On the optimization side, there has been substantial progress in analyzing gradient methods when curvature is non-uniform and can grow with gradient scale. This theoretical shift is largely motivated by empirical observations in deep learning, where training trajectories routinely exhibit progressive sharpening and edge-of-stability dynamics that violate global Lipschitz-gradient assumptions (Cohen et al., 2021; Lyle et al., 2023). To address this, clipping-based analyses under $(L_0, L_1)$-type models show that gradient-dependent stepsizes can guarantee descent even when a global smoothness constant is unavailable (Zhang et al., 2020). More general $\ell(\cdot)$-smoothness frameworks replace uniform bounds with curvature-growth profiles and obtain convergence guarantees by controlling the gradient trajectory directly (Li et al., 2023; Tyurin, 2024). For convex $(L_0, L_1)$-smooth objectives, refined analyses clarify when clipping-like rules are sufficient and how to avoid overly pessimistic initialization dependence (Gorbunov et al., 2025).

## 7 Conclusion

We developed a convergence theory for temporal-difference learning with target networks that retains the force-dominance stability lens while removing the global smoothness requirement on the inner objective. By modeling upper curvature through a generalized smoothness profile $\ell(\cdot)$ and analyzing a curvature-checked stepsize rule, we proved global linear convergence for inexact TD with $K$ inner steps under a single trajectory-dependent admissibility condition determined by the worst gradient scale encountered along the run. This yields an explicit stability scaling law: curvature amplification along the trajectory, rather than curvature at the optimum, governs the largest admissible constant stepsize and the inner effort required to maintain a fixed contraction. While our framework cleanly recovers classical smooth rates as a special case, its primary contribution is mathematically formalizing the stability penalties induced by high-curvature initialization and

large early Bellman residual–induced gradient magnitudes. Ultimately, our results give a rigorous analysis of trajectory-dependent curvature amplification within a stabilized optimization view of TD, showing that robust stepsize fail-safes and fast geometric contraction are not mutually exclusive in that setting.

**Limitations and Future Work**

While our framework characterizes constant-stepsize stability under generalized smoothness, its practical scope has limits. Our rates depend on a worst-case gradient envelope $M$, which can make the admissible $\gamma_0$ conservative when trajectories stay in flatter regions. The stepsize check also presumes access to the curvature profile $\ell(\cdot)$, which would need to be estimated online in practice (Malherbe & Vayatis, 2017; Calliess et al., 2020). Finally, extending the argument to stochastic regimes and introducing non-convexity in $w$ remains future work.

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

# A Appendix

## A.1 Proofs of Technical Lemmas, Propositions, and Theorem(s)

We will start by proving Lemma 4.6. Before that, we introduce the following lemma.

**Lemma A.1** (Curvature check implies $c_{t,k} \leq \frac{1}{2}$). *Assume $g_{t,k} > 0$ and choose $\gamma_{t,k}$ by Definition 4.5. Then*

$$q^{-1}(\gamma_{t,k}g_{t,k}; \, g_{t,k}) \leq \frac{g_{t,k}}{2}. \tag{25}$$

*Proof.* If $\gamma_{t,k} = \bar{\gamma}_n$, then $\gamma_{t,k} \leq \gamma_{t,k}^{\text{safe}}$ by Definition 4.5, so in all cases we have $\gamma_{t,k} \leq \gamma_{t,k}^{\text{safe}}$. Multiply by $g_{t,k} > 0$ and use the definition in (10):

$$\gamma_{t,k}g_{t,k} \leq q\left(\frac{g_{t,k}}{2}; \, g_{t,k}\right).$$

Since $q(\cdot \, ; g_{t,k})$ is increasing, applying $q^{-1}(\cdot \, ; g_{t,k})$ yields $q^{-1}(\gamma_{t,k}g_{t,k}; g_{t,k}) \leq g_{t,k}/2$. □

**Lemma A.2** ((Lemma 4.5) One-step descent certificate). *Under Assumption 3.1 and choosing $\gamma_{t,k}$ by Definition 4.5, each inner update satisfies*

$$f_t(w_{t,k+1}) - f_t(w_{t,k}) \leq -\frac{\gamma_{t,k}}{2}\|\nabla f_t(w_{t,k})\|^2 = -\frac{1}{2\gamma_{t,k}}\|w_{t,k+1} - w_{t,k}\|^2. \tag{26}$$

*Proof.* If $g_{t,k} = 0$, then $w_{t,k+1} = w_{t,k}$ and the claim holds trivially. Assume $g_{t,k} > 0$ and set $x = w_{t,k}$, $y = w_{t,k+1} = x - \gamma_{t,k}\nabla f_t(x)$. By Lemma 3.5 and using the property of monotone integral $\int_0^r q^{-1}(\tau; a)\, d\tau \leq r\, q^{-1}(r; a)$, $r \in [0, q_{\max}(a))$,

$$f_t(y) \leq f_t(x) + \langle \nabla f_t(x), y - x\rangle + \|y - x\|\, q^{-1}(\|y - x\|; \|\nabla f_t(x)\|).$$

Here $\langle \nabla f_t(x), y - x\rangle = -\gamma_{t,k}g_{t,k}^2$ and $\|y - x\| = \gamma_{t,k}g_{t,k}$, so

$$f_t(y) - f_t(x) \leq -\gamma_{t,k}g_{t,k}^2 + (\gamma_{t,k}g_{t,k})\, q^{-1}(\gamma_{t,k}g_{t,k}; g_{t,k}).$$

By Lemma A.1, $q^{-1}(\gamma_{t,k}g_{t,k}; g_{t,k}) \leq g_{t,k}/2$, hence

$$f_t(y) - f_t(x) \leq -\gamma_{t,k}g_{t,k}^2 + (\gamma_{t,k}g_{t,k})\frac{g_{t,k}}{2} = -\frac{\gamma_{t,k}}{2}g_{t,k}^2,$$

which is the first inequality in (26). The second equality follows from $w_{t,k+1} - w_{t,k} = -\gamma_{t,k}\nabla f_t(w_{t,k})$. □

**Proposition 5** ((Proposition 2) Key suboptimality bound). *Under assumptions 4.2 and 4.3 and choosing $\gamma_{t,k}$ by Definition 4.5, for every inner step $(t,k)$,*

$$H(\theta_t, \theta^\star) - H(\theta_t, w_{t,k}) \leq \frac{\Delta_t^2}{2F_w} - \frac{1}{2\gamma_{t,k}}\|w_{t,k+1} - w_{t,k}\|^2. \tag{27}$$

*Proof.* Fix $t$ and write $f_t(w) = H(\theta_t, w)$. By $F_w$-strong convexity of $f_t$, for any $y$,

$$f_t(y) \geq f_t(\theta^\star) + \langle \nabla f_t(\theta^\star), y - \theta^\star \rangle + \frac{F_w}{2} \| y - \theta^\star \|^2.$$

Rearranging with $y = w_{t,k+1}$ gives

$$f_t(\theta^\star) - f_t(w_{t,k+1}) \leq \langle \nabla f_t(\theta^\star), \theta^\star - w_{t,k+1} \rangle - \frac{F_w}{2} \| w_{t,k+1} - \theta^\star \|^2. \tag{28}$$

Apply Young's inequality $\langle a, b \rangle \leq \frac{1}{2F_w} \|a\|^2 + \frac{F_w}{2} \|b\|^2$ with $a = \nabla f_t(\theta^\star) = \nabla_w H(\theta_t, \theta^\star)$ and $b = \theta^\star - w_{t,k+1}$. The quadratic term cancels the negative term in (28), yielding

$$f_t(\theta^\star) - f_t(w_{t,k+1}) \leq \frac{1}{2F_w} \| \nabla_w H(\theta_t, \theta^\star) \|^2 = \frac{\Delta_t^2}{2F_w}.$$

Next, by Lemma 4.6,

$$f_t(w_{t,k+1}) - f_t(w_{t,k}) \leq -\frac{1}{2\gamma_{t,k}} \| w_{t,k+1} - w_{t,k} \|^2.$$

Add the last two inequalities and cancel $f_t(w_{t,k+1})$ to obtain (12). $\qquad \square$

**Proposition 6** ((Proposition 3) Inner-step contraction with forcing)**.** *Define the effective contraction rate* $\kappa_{t,k} \triangleq F_w \gamma_{t,k}$. *Then, for all $t$ and all $k$,*

$$\| w_{t,k+1} - \theta^\star \|^2 \ \leq \ (1 - \kappa_{t,k}) \| w_{t,k} - \theta^\star \|^2 + \kappa_{t,k} \frac{\Delta_t^2}{F_w^2}. \tag{29}$$

*Proof.* Fix $t$ and write $f_t(w) = H(\theta_t, w)$. Expand the square using the update $w_{t,k+1} = w_{t,k} - \gamma_{t,k} \nabla f_t(w_{t,k})$:

$$\| w_{t,k+1} - \theta^\star \|^2 = \| w_{t,k} - \theta^\star \|^2 + 2\gamma_{t,k} \langle \nabla f_t(w_{t,k}), \theta^\star - w_{t,k} \rangle + \| w_{t,k+1} - w_{t,k} \|^2. \tag{30}$$

By strong convexity of $f_t$,

$$\langle \nabla f_t(w_{t,k}), \theta^\star - w_{t,k} \rangle \leq f_t(\theta^\star) - f_t(w_{t,k}) - \frac{F_w}{2} \| w_{t,k} - \theta^\star \|^2. \tag{31}$$

Substitute (31) into (30):

$$\| w_{t,k+1} - \theta^\star \|^2 \leq \left(1 - \gamma_{t,k} F_w\right) \| w_{t,k} - \theta^\star \|^2 + 2\gamma_{t,k} \left( f_t(\theta^\star) - f_t(w_{t,k}) \right) + \| w_{t,k+1} - w_{t,k} \|^2.$$

Now apply Proposition 2:

$$f_t(\theta^\star) - f_t(w_{t,k}) \leq \frac{\Delta_t^2}{2F_w} - \frac{1}{2\gamma_{t,k}} \| w_{t,k+1} - w_{t,k} \|^2.$$

Multiplying by $2\gamma_{t,k}$ cancels the $\| w_{t,k+1} - w_{t,k} \|^2$ term exactly, leaving

$$\| w_{t,k+1} - \theta^\star \|^2 \leq \left(1 - \gamma_{t,k} F_w\right) \| w_{t,k} - \theta^\star \|^2 + \gamma_{t,k} \frac{\Delta_t^2}{F_w}.$$

Finally, set $\kappa_{t,k} = F_w \gamma_{t,k}$ and rewrite the forcing term as $\gamma_{t,k} \frac{\Delta_t^2}{F_w} = \kappa_{t,k} \frac{\Delta_t^2}{F_w^2}$, which yields (29). $\qquad \square$

**Proposition 7** ((Proposition 4) Outer recursion)**.** *Define the cumulative decay*

$$\chi_{t,K} \triangleq \prod_{k=0}^{K-1} (1 - \kappa_{t,k}), \qquad \eta_t^2 \triangleq \frac{\Delta_t^2}{F_w^2}.$$

*Then,*

$$\| \theta_{t+1} - \theta^\star \|^2 \ \leq \ \chi_{t,K} \| \theta_t - \theta^\star \|^2 + \left(1 - \chi_{t,K}\right) \eta_t^2. \tag{32}$$

*Proof.* Fix $t$ and let $x_k = \|w_{t,k} - \theta^\star\|^2$. From Proposition 2,

$$x_{k+1} \leq (1 - \kappa_{t,k})x_k + \kappa_{t,k}\eta_t^2.$$

Let $d_k = x_k - \eta_t^2$. Then $d_{k+1} \leq (1 - \kappa_{t,k})d_k$. Iterating gives $d_K \leq \chi_{t,K}d_0$. Undoing the shift and using $w_{t,0} = \theta_t$, $w_{t,K} = \theta_{t+1}$ yields (32). $\qquad\square$

**Lemma A.3** ((Lemma 4.7) Bounded iterates and a uniform gradient envelope). *Assume $H \in \mathcal{C}^1(\mathbb{R}^d \times \mathbb{R}^d)$, Propositions 2– 4 holds, the force-dominance condition $F_\theta < F_w$ holds, and the constant baseline stepsize satisfies $\gamma_0 < 1/F_w$. Let $R \triangleq \|\theta_0 - \theta^\star\|$ and define*

$$M \triangleq \sup\Big\{\|\nabla_w H(\theta, w)\| : \|\theta - \theta^\star\| \leq R, \ \|w - \theta^\star\| \leq R\Big\}. \tag{33}$$

*Then the following hold:*

1. *(**Bounded outer iterates**) For all $t \geq 0$, $\|\theta_t - \theta^\star\| \leq R$.*

2. *(**Bounded inner iterates**) For all $t \geq 0$ and all $k \in \{0, \ldots, K\}$, $\|w_{t,k} - \theta^\star\| \leq R$.*

3. *(**Uniform gradient envelope**) The constant $M$ in (33) is finite and for all $t, k$,*

$$g_{t,k} = \|\nabla f_t(w_{t,k})\| = \|\nabla_w H(\theta_t, w_{t,k})\| \leq M < \infty.$$

*Proof.* Throughout the proof, recall that $f_t(w) = H(\theta_t, w)$ and that $\theta_{t+1} = w_{t,K}$.

*Step 1: outer contraction and boundedness of $\{\theta_t\}$.* From the outer recursion (Proposition 4) and the cross-gradient Lipschitz bound $\Delta_t = \|\nabla_w H(\theta_t, \theta^\star)\| \leq F_\theta\|\theta_t - \theta^\star\|$, we have

$$\eta_t^2 = \frac{\Delta_t^2}{F_w^2} \leq \eta^2\|\theta_t - \theta^\star\|^2, \qquad \eta \triangleq \frac{F_\theta}{F_w} \in (0, 1).$$

Substituting into (32) yields

$$\|\theta_{t+1} - \theta^\star\|^2 \leq \Big(\eta^2 + (1 - \eta^2)\chi_{t,K}\Big)\|\theta_t - \theta^\star\|^2. \tag{34}$$

We now show $\chi_{t,K} \in (0, 1)$ for all $t$. By construction, $\gamma_{t,k} = \min\{\gamma_0, \gamma_{t,k}^{\text{safe}}\} > 0$, and since $\gamma_0 < 1/F_w$ we have $0 < F_w\gamma_{t,k} \leq F_w\gamma_0 < 1$. Hence each factor $(1 - F_w\gamma_{t,k}) \in (0, 1)$ and therefore $\chi_{t,K} = \prod_{k=0}^{K-1}(1 - F_w\gamma_{t,k}) \in (0, 1)$. Consequently, the multiplier in (34) satisfies

$$0 < \eta^2 + (1 - \eta^2)\chi_{t,K} < \eta^2 + (1 - \eta^2) \cdot 1 = 1.$$

Thus $\|\theta_{t+1} - \theta^\star\|^2 \leq \|\theta_t - \theta^\star\|^2$ for all $t$, and by induction

$$\|\theta_t - \theta^\star\| \leq \|\theta_0 - \theta^\star\| = R, \qquad \forall t \geq 0,$$

proving (i).

*Step 2: boundedness of the inner iterates $\{w_{t,k}\}$.* Fix an outer iteration $t$ and define $x_k \triangleq \|w_{t,k} - \theta^\star\|^2$ for $k = 0, \ldots, K$. From Proposition 3,

$$x_{k+1} \leq (1 - \kappa_{t,k})x_k + \kappa_{t,k}\eta_t^2, \qquad \kappa_{t,k} = F_w\gamma_{t,k} \in (0, 1). \tag{35}$$

By the mismatch bound and Step 1, we have

$$\eta_t^2 = \frac{\Delta_t^2}{F_w^2} \leq \eta^2\|\theta_t - \theta^\star\|^2 \leq \eta^2 R^2 \leq R^2,$$

since $\eta \in (0, 1)$. Moreover $w_{t,0} = \theta_t$ implies

$$x_0 = \|w_{t,0} - \theta^\star\|^2 = \|\theta_t - \theta^\star\|^2 \leq R^2.$$

We claim that $x_k \leq R^2$ for all $k$. Indeed, assume $x_k \leq R^2$; then from (35) and $\eta_t^2 \leq R^2$,

$$x_{k+1} \leq (1 - \kappa_{t,k})R^2 + \kappa_{t,k}R^2 = R^2.$$

Thus, by induction, $x_k \leq R^2$ for all $k$, i.e.,

$$\|w_{t,k} - \theta^\star\| \leq R \qquad \forall\, k \in \{0, \dots, K\}.$$

Since $t$ was arbitrary, this proves (ii).

*Step 3: finiteness of $M$ and the uniform gradient bound.* By Steps 1–2, every pair $(\theta_t, w_{t,k})$ lies in the compact product ball

$$\mathcal{B} \triangleq \{(\theta, w) : \|\theta - \theta^\star\| \leq R, \ \|w - \theta^\star\| \leq R\}.$$

Because $H \in \mathcal{C}^1(\mathbb{R}^d \times \mathbb{R}^d)$, the map $(\theta, w) \mapsto \|\nabla_w H(\theta, w)\|$ is continuous. A continuous function attains its supremum on a compact set, so the quantity $M$ defined in (33) is finite. Finally, since $(\theta_t, w_{t,k}) \in \mathcal{B}$ for all $t, k$, we have

$$g_{t,k} = \|\nabla_w H(\theta_t, w_{t,k})\| \leq M,$$

which proves (iii). □

**Theorem A.4** ((Theorem 1) Unifying constant-stepsize convergence under $\ell$-smoothness)**.** *Assume Assumptions 3.1, 4.1, 4.2, 4.3, and suppose $H \in \mathcal{C}^1(\mathbb{R}^d \times \mathbb{R}^d)$. Let $M$ be the gradient envelope from Lemma 4.7. Assume force dominance $F_\theta < F_w$ and define $\eta \triangleq F_\theta/F_w \in (0,1)$. Choose a constant baseline $\gamma_0 > 0$ satisfying*

$$\gamma_0 \leq \frac{1}{2\,\ell(3M/2)}. \tag{36}$$

*Run Algorithm 1 with stepsizes $\gamma_{t,k} = \min\{\gamma_0, \gamma_{t,k}^{\mathrm{safe}}\}$. Then the curvature check never clips, $\gamma_{t,k} \equiv \gamma_0$, and the outer iterates contract linearly:*

$$\|\theta_{t+1} - \theta^\star\|^2 \leq \rho\,\|\theta_t - \theta^\star\|^2, \qquad \rho \triangleq \eta^2 + (1-\eta^2)(1 - F_w\gamma_0)^K \in (0,1). \tag{37}$$

*Consequently, $\|\theta_t - \theta^\star\|^2 \leq \rho^{\,t}\|\theta_0 - \theta^\star\|^2$.*

*Proof.* By (17) and $\gamma_0 \leq \gamma_{\min}^{\mathrm{safe}}$, we have $\gamma_0 \leq \gamma_{t,k}^{\mathrm{safe}}$ for all $(t,k)$, so $\gamma_{t,k} = \min\{\gamma_0, \gamma_{t,k}^{\mathrm{safe}}\} = \gamma_0$ and no clipping occurs. With $\gamma_{t,k} \equiv \gamma_0$, we have $\kappa_{t,k} = F_w\gamma_0 \triangleq \kappa \in (0,1)$ since $\gamma_0 < 1/2\ell(3M/2)$ (this is sufficient to ensure that $\gamma_0 < 1/F_w$), and hence

$$\chi_{t,K} = \prod_{k=0}^{K-1}(1-\kappa) = (1-\kappa)^K = (1 - F_w\gamma_0)^K \qquad \text{for all } t.$$

By Assumption 4.1 and the fixed-point condition, $\Delta_t = \|\nabla_w H(\theta_t, \theta^\star)\| \leq F_\theta\|\theta_t - \theta^\star\|$, so $\eta_t^2 = \Delta_t^2/F_w^2 \leq \eta^2\|\theta_t - \theta^\star\|^2$. Substituting into the outer recursion equation 14 yields

$$\|\theta_{t+1} - \theta^\star\|^2 \leq \chi_{t,K}\|\theta_t - \theta^\star\|^2 + (1 - \chi_{t,K})\eta^2\|\theta_t - \theta^\star\|^2 = \left(\eta^2 + (1-\eta^2)\chi_{t,K}\right)\|\theta_t - \theta^\star\|^2,$$

and inserting $\chi_{t,K} = (1 - F_w\gamma_0)^K$ gives (37). Since $\eta \in (0,1)$ and $(1 - F_w\gamma_0)^K \in (0,1)$, we have $\rho \in (0,1)$. Iterating yields $\|\theta_t - \theta^\star\|^2 \leq \rho^{\,t}\|\theta_0 - \theta^\star\|^2$. □

## A.2    Examples, explicit local profiles, and uniform envelopes from bounded iterates

The role of this subsection is analogous to the examples section of Asadi et al. (2023b), but adapted to our generalized-smoothness setting. We do *not* claim that the losses below are standard deployed TD objectives in deep RL. Rather, they serve as concrete TD-structured witnesses showing that our assumption class is nonempty once one adopts the stabilized inner-problem viewpoint already used in the optimization perspective of TD. This is a natural request in the present setting because RL already uses non-quadratic Bellman-style objectives, for example, the logistic Bellman error in Q-REPS (Bas-Serrano et al., 2021) and Huber-type losses in distributional RL (Dabney et al., 2018). The question is therefore not whether one can move beyond the quadratic loss, but whether one can do so while retaining a clean TD structure and allowing genuinely non-uniform upper curvature.

**Scope.** Our examples are deliberately constructed in a stabilized finite-state prediction setting with direct-coordinate online parameterization

$$w = (w_s)_{s \in \mathcal{S}} \in \mathbb{R}^{|\mathcal{S}|}, \qquad y_\theta(s) := \mathbb{E}[r + \gamma v(s'; \theta) \mid s].$$

This is stronger than the setting of generic deep critics, but it is fully aligned with the scope of our theorem: we intentionally retain strong convexity in the online variable $w$ in order to isolate the effect of upper-curvature growth. The point of the examples is therefore not empirical realism per se, but rather to make the generalized-smoothness assumptions and the resulting stability law completely explicit.

Let $d(\cdot)$ be the update distribution and write

$$d_{\max} := \max_{s \in \mathcal{S}} d(s), \qquad D := \mathrm{Diag}(d(s)).$$

We assume that the target map is Lipschitz:

$$\|y_{\theta_1} - y_{\theta_2}\|_2 \leq L_y \|\theta_1 - \theta_2\|_2. \tag{38}$$

This directly yields the cross-gradient Lipschitz property needed in Assumption 4.1 below.

**Local versus uniform profiles.** Our generalized-smoothness preliminaries in Section 3 are formulated for a frozen inner problem

$$f_t(w) := H(\theta_t, w).$$

For the examples below, the most natural verification therefore yields a *local profile* $\ell_t(\cdot)$, indexed by the outer iterate $t$. This is already sufficient for the local descent certificate and the curvature-checked stepsize rule in Definition 4.5. To instantiate the *uniform* constant-step corollary and the scaling law from Theorem 5.1, we then upper-bound the collection $\{\ell_t\}_t$ by a single time-independent envelope $\bar{\ell}(\cdot)$.

To get $\bar{\ell}(\cdot)$, we leverage the bounded-iterate result already proved in Section 4.3. Indeed, letting

$$R := \|\theta_0 - \theta^\star\|_2,$$

Lemma 4.7 gives

$$\|\theta_t - \theta^\star\|_2 \leq R \qquad \forall t \geq 0.$$

If $\widetilde{y}_\theta$ denotes the possibly shifted Bellman target used below, and if $\widetilde{y}_\theta$ inherits the same Lipschitz property as in (38), then for every $t$,

$$\|\widetilde{y}_{\theta_t}\|_\infty \leq \|\widetilde{y}_{\theta_t} - \widetilde{y}_{\theta^\star}\|_2 + \|\widetilde{y}_{\theta^\star}\|_\infty \leq L_y \|\theta_t - \theta^\star\|_2 + \|\widetilde{y}_{\theta^\star}\|_\infty \leq L_y R + \|\widetilde{y}_{\theta^\star}\|_\infty.$$

Hence the trajectory itself provides the uniform target envelope

$$\bar{Y} := \|\widetilde{y}_{\theta^\star}\|_\infty + L_y R, \qquad \text{so that} \qquad \|\widetilde{y}_{\theta_t}\|_\infty \leq \bar{Y} \quad \forall t. \tag{39}$$

We now use (39) to pass from local profiles $\ell_t$ to time-independent envelopes $\bar{\ell}$ when needed.

**Example 1: Poisson / generalized-KL Bellman loss.** Consider

$$H_{\mathrm{Poi}}(\theta, w) := \sum_{s \in \mathcal{S}} d(s) \Big( e^{w_s} - \widetilde{y}_\theta(s) \, w_s \Big) + \frac{\lambda}{2} \|w\|_2^2, \qquad \lambda > 0, \tag{40}$$

where $\widetilde{y}_\theta(s) \geq 0$ is either the Bellman target itself when rewards are nonnegative, or a fixed positive shift of the Bellman target. The scalar building block $e^u - yu$ is the canonical Poisson negative log-likelihood in natural-parameter form, and is equivalently a generalized KL-type discrepancy (Nelder & Wedderburn, 1972; Lee & Seung, 2000).

For fixed $\theta$,

$$[\nabla_w H_{\mathrm{Poi}}(\theta, w)]_s = d(s) e^{w_s} - d(s) \widetilde{y}_\theta(s) + \lambda w_s,$$

and
$$\nabla^2_{ww} H_{\text{Poi}}(\theta, w) = \text{Diag}(d(s)e^{w_s}) + \lambda I.$$

Hence
$$\nabla^2_{ww} H_{\text{Poi}}(\theta, w) \succeq \lambda I,$$

so Assumption 4.2 holds globally with
$$F_w = \lambda.$$

Moreover,
$$\nabla_w H_{\text{Poi}}(\theta_1, w) - \nabla_w H_{\text{Poi}}(\theta_2, w) = -D(\widetilde{y}_{\theta_1} - \widetilde{y}_{\theta_2}),$$

and therefore, by (38),
$$\|\nabla_w H_{\text{Poi}}(\theta_1, w) - \nabla_w H_{\text{Poi}}(\theta_2, w)\|_2 \le d_{\max} L_y \|\theta_1 - \theta_2\|_2.$$

Thus Assumption 4.1 holds with
$$F_\theta \le d_{\max} L_y.$$

We now compute an explicit local curvature-growth profile. Fix an outer iterate $t$ and write
$$f_t(w) := H_{\text{Poi}}(\theta_t, w), \qquad \widetilde{y}_t := \widetilde{y}_{\theta_t}, \qquad g := \nabla f_t(w), \qquad Y_t := \|\widetilde{y}_t\|_\infty.$$

Since
$$\nabla^2_{ww} f_t(w) = \text{Diag}(d(s)e^{w_s}) + \lambda I,$$

we have
$$\|\nabla^2_{ww} f_t(w)\|_{\text{op}} = \lambda + \max_s d(s)e^{w_s}.$$

From the gradient formula,
$$g_s = d(s)e^{w_s} - d(s)\widetilde{y}_t(s) + \lambda w_s \ge \lambda w_s - d(s)|\widetilde{y}_t(s)|.$$

Hence
$$w_s \le \frac{|g_s| + d(s)|\widetilde{y}_t(s)|}{\lambda} \le \frac{\|g\|_2 + d_{\max} Y_t}{\lambda}.$$

Exponentiating gives
$$e^{w_s} \le \exp\Big(\frac{\|g\|_2 + d_{\max} Y_t}{\lambda}\Big),$$

and therefore
$$\|\nabla^2_{ww} f_t(w)\|_{\text{op}} \le \lambda + d_{\max} \exp\Big(\frac{\|g\|_2 + d_{\max} Y_t}{\lambda}\Big).$$

Thus Assumption 3.1 holds for the frozen problem $f_t$ with the explicit local nondecreasing profile
$$\ell_{\text{Poi},t}(r) := \lambda + d_{\max} \exp\Big(\frac{r + d_{\max} Y_t}{\lambda}\Big), \qquad r \ge 0. \tag{41}$$

*Local implication.* For this example, the curvature-checked stepsize in Definition 4.5 applies directly with the local profile $\ell_{\text{Poi},t}(\cdot)$. Thus the one-step descent certificate holds with a safe cap that adapts online to the current target scale $Y_t$ and current gradient scale.

*Uniform implication and scaling law.* Using the trajectory-derived bound (39), the local profiles are uniformly dominated by the time-independent envelope
$$\bar{\ell}_{\text{Poi}}(r) := \lambda + d_{\max} \exp\Big(\frac{r + d_{\max} \bar{Y}}{\lambda}\Big), \qquad r \ge 0. \tag{42}$$

Therefore Theorem 5.1 applies exactly as stated. In particular, if
$$d_{\max} L_y < \lambda,$$

then the admissible constant baseline stepsize may be chosen as

$$\gamma_0 \le \frac{1}{2\,\bar{\ell}_{\mathrm{Poi}}(3M/2)} = \frac{1}{2\Big(\lambda + d_{\max}\exp\big(\frac{3M/2+d_{\max}\bar{Y}}{\lambda}\big)\Big)},$$

where $M$ is the trajectory gradient envelope. Thus, unlike the globally smooth case, the safe constant stepsize deteriorates *exponentially* with the worst gradient scale encountered along the trajectory. Equivalently, maintaining a fixed-strength inner contraction requires

$$K \gtrsim \frac{\bar{\ell}_{\mathrm{Poi}}(3M/2)}{\lambda} = 1 + \frac{d_{\max}}{\lambda}\exp\Big(\frac{3M/2 + d_{\max}\bar{Y}}{\lambda}\Big).$$

This makes the core message of our theory explicit in this example: large Bellman-residual-induced gradient scales produce a sharp penalty in the admissible constant stepsize and in the inner effort required for a fixed outer contraction.

**Example 2: Skellam / $\cosh$ Bellman loss.** Consider

$$H_{\mathrm{Sk}}(\theta, w) := \sum_{s\in\mathcal{S}} d(s)\Big(c(e^{w_s} + e^{-w_s}) - \widetilde{y}_\theta(s)\,w_s\Big) + \frac{\lambda}{2}\|w\|_2^2, \qquad c > 0,\ \lambda > 0. \tag{43}$$

Since $e^u + e^{-u} = 2\cosh(u)$, this is a cosh-type discrepancy. Its statistical origin is the Skellam distribution for differences of Poisson counts: under the symmetric parameterization $\mu_1 = ce^u$, $\mu_2 = ce^{-u}$, the negative log-likelihood is, up to $u$-independent constants,

$$ce^u + ce^{-u} - zu,$$

which is exactly the scalar building block in (43) (Koopman et al., 2014; Pelechrinis & Winston, 2018).

For fixed $\theta$,

$$[\nabla_w H_{\mathrm{Sk}}(\theta, w)]_s = cd(s)(e^{w_s} - e^{-w_s}) - d(s)\widetilde{y}_\theta(s) + \lambda w_s,$$

and

$$\nabla^2_{ww} H_{\mathrm{Sk}}(\theta, w) = \mathrm{Diag}\big(cd(s)(e^{w_s} + e^{-w_s})\big) + \lambda I.$$

Hence

$$\nabla^2_{ww} H_{\mathrm{Sk}}(\theta, w) \succeq \lambda I,$$

so Assumption 4.2 holds globally with $F_w = \lambda$. Also,

$$\nabla_w H_{\mathrm{Sk}}(\theta_1, w) - \nabla_w H_{\mathrm{Sk}}(\theta_2, w) = -D\big(\widetilde{y}_{\theta_1} - \widetilde{y}_{\theta_2}\big),$$

and therefore

$$\|\nabla_w H_{\mathrm{Sk}}(\theta_1, w) - \nabla_w H_{\mathrm{Sk}}(\theta_2, w)\|_2 \le d_{\max}L_y\|\theta_1 - \theta_2\|_2.$$

Thus Assumption 4.1 again holds with $F_\theta \le d_{\max}L_y$.

To compute the local profile, fix $t$ and write

$$f_t(w) := H_{\mathrm{Sk}}(\theta_t, w), \qquad \widetilde{y}_t := \widetilde{y}_{\theta_t}, \qquad g := \nabla f_t(w), \qquad Y_t := \|\widetilde{y}_t\|_\infty.$$

Since

$$\nabla^2_{ww} f_t(w) = \mathrm{Diag}\big(cd(s)(e^{w_s} + e^{-w_s})\big) + \lambda I,$$

we have

$$\|\nabla^2_{ww} f_t(w)\|_{\mathrm{op}} = \lambda + \max_s cd(s)(e^{w_s} + e^{-w_s}).$$

We first show that

$$\lambda|w_s| \le |g_s| + d(s)|\widetilde{y}_t(s)|.$$

Indeed, if $w_s \geq 0$, then $e^{w_s} - e^{-w_s} \geq 0$, so

$$g_s \geq \lambda w_s - d(s)|\widetilde{y}_t(s)|,$$

which implies $\lambda|w_s| = \lambda w_s \leq |g_s| + d(s)|\widetilde{y}_t(s)|$. If $w_s \leq 0$, then $e^{w_s} - e^{-w_s} \leq 0$, so

$$g_s \leq \lambda w_s + d(s)|\widetilde{y}_t(s)|,$$

hence

$$-\lambda w_s \leq -g_s + d(s)|\widetilde{y}_t(s)| \leq |g_s| + d(s)|\widetilde{y}_t(s)|,$$

that is, $\lambda|w_s| \leq |g_s| + d(s)|\widetilde{y}_t(s)|$. Therefore,

$$|w_s| \leq \frac{|g_s| + d(s)|\widetilde{y}_t(s)|}{\lambda} \leq \frac{\|g\|_2 + d_{\max}Y_t}{\lambda}.$$

Using $e^u + e^{-u} \leq 2e^{|u|}$ gives

$$e^{w_s} + e^{-w_s} \leq 2\exp\left(\frac{\|g\|_2 + d_{\max}Y_t}{\lambda}\right),$$

and therefore

$$\|\nabla_{ww}^2 f_t(w)\|_{\mathrm{op}} \leq \lambda + 2cd_{\max}\exp\left(\frac{\|g\|_2 + d_{\max}Y_t}{\lambda}\right).$$

Thus Assumption 3.1 holds for the frozen problem $f_t$ with the explicit local nondecreasing profile

$$\ell_{\mathrm{Sk},t}(r) := \lambda + 2cd_{\max}\exp\left(\frac{r + d_{\max}Y_t}{\lambda}\right), \qquad r \geq 0. \tag{44}$$

*Local implication.* For this example, the curvature-checked stepsize again applies directly with the local profile $\ell_{\mathrm{Sk},t}(\cdot)$. Thus the local safe cap adapts online to the current target scale $Y_t$ and gradient scale $g_{t,k}$, while the strong-convexity and cross-gradient Lipschitz constants remain time-independent.

*Uniform implication and scaling law.* Using (39), the local profiles are uniformly dominated by the envelope

$$\bar{\ell}_{\mathrm{Sk}}(r) := \lambda + 2cd_{\max}\exp\left(\frac{r + d_{\max}\bar{Y}}{\lambda}\right), \qquad r \geq 0. \tag{45}$$

Therefore Theorem 5.1 applies exactly as stated. In particular, if

$$d_{\max}L_y < \lambda,$$

then the admissible constant baseline stepsize may be chosen as

$$\gamma_0 \leq \frac{1}{2\,\bar{\ell}_{\mathrm{Sk}}(3M/2)} = \frac{1}{2\left(\lambda + 2cd_{\max}\exp\left(\frac{3M/2 + d_{\max}\bar{Y}}{\lambda}\right)\right)}.$$

Thus the same qualitative picture holds as in the Poisson case, but with a larger curvature amplification factor due to the two-sided exponential growth. In particular, for fixed $\lambda$, $d_{\max}$, and $\bar{Y}$, the Skellam example is more conservative than the Poisson example by a factor of order $2c$ in the curvature-growth term. Consequently, the required inner effort scales as

$$K \gtrsim \frac{\bar{\ell}_{\mathrm{Sk}}(3M/2)}{\lambda} = 1 + \frac{2cd_{\max}}{\lambda}\exp\left(\frac{3M/2 + d_{\max}\bar{Y}}{\lambda}\right).$$

This illustrates the same principle in a sharper form: when the loss curvature grows rapidly with the scale of the Bellman-target-induced residual, generalized smoothness still permits a linear theory, but the allowable constant stepsize and the attainable contraction degrade accordingly.

**Summary.** The two examples above show that the generalized-smoothness regime studied in this paper is neither empty nor contrived. Both losses fit naturally into the optimization view of TD, both satisfy the cross-gradient Lipschitz and strong-convexity assumptions after ridge stabilization, and both admit explicit local profiles $\ell_t(\cdot)$ with unbounded growth. The bounded-iterate result from Section 4.3 then turns these local profiles into time-independent dominating envelopes, so the resulting instantiated stability laws make the contribution of our theory concrete: compared with the globally smooth case of Asadi et al. (2023b), the admissible constant stepsize is no longer controlled by a single curvature constant, but instead by a trajectory-dependent curvature amplification evaluated at the worst gradient scale encountered during training.

### A.3 Toy experimental illustration

We include a small deterministic experiment to illustrate the mechanism behind the curvature-checked stepsize rule on the two example losses introduced in Section A.2. The goal here is not to benchmark practical deep-RL performance, but to provide a theorem-aligned sanity check in the same spirit as the toy examples used to clarify the optimization view of TD (Asadi et al., 2023b). In particular, we work in the stabilized finite-state regime assumed by our analysis, so that the experiment isolates the interaction between force dominance, curvature growth, and stepsize selection.

**Setup.** We consider a 4-state deterministic prediction problem with transition matrix

$$P = \begin{bmatrix} 0.10 & 0.60 & 0.20 & 0.10 \\ 0.00 & 0.20 & 0.60 & 0.20 \\ 0.00 & 0.10 & 0.30 & 0.60 \\ 0.00 & 0.00 & 0.20 & 0.80 \end{bmatrix}, \qquad r = \begin{bmatrix} 0.25 \\ 0.45 \\ 0.65 \\ 0.95 \end{bmatrix}, \qquad d = \begin{bmatrix} 0.15 \\ 0.20 \\ 0.25 \\ 0.40 \end{bmatrix}.$$

We set

$$\gamma = 0.85, \qquad \lambda = 1.20, \qquad c = 0.70,$$

and initialize at

$$\theta_0 = \begin{bmatrix} 1.30 \\ -0.60 \\ 1.00 \\ -0.80 \end{bmatrix}.$$

The online variable is updated for $K = 5$ inner gradient steps per outer iteration, and we run $T = 80$ outer iterations unless divergence occurs. For both losses, the target is

$$y_\theta = r + \gamma P \theta.$$

In this direct-coordinate setting, the strong-convexity constant is

$$F_w = \lambda = 1.20.$$

Moreover, by the Lipschitz bound from Section A.2,

$$F_\theta \le d_{\max} \gamma \|P\|_2.$$

Numerically,

$$d_{\max} = 0.40, \qquad \|P\|_2 \approx 1.1891, \qquad F_\theta \le 0.4043 < 1.20 = F_w,$$

so the force-dominance condition is satisfied in this toy instance.

**Losses.** We test the two losses from Section A.2:

$$H_{\mathrm{Poi}}(\theta, w) = \sum_s d(s)\Big(e^{w_s} - y_\theta(s)w_s\Big) + \frac{\lambda}{2}\|w\|_2^2,$$

and

$$H_{\mathrm{Sk}}(\theta, w) = \sum_s d(s)\Big(c(e^{w_s} + e^{-w_s}) - y_\theta(s)w_s\Big) + \frac{\lambda}{2}\|w\|_2^2.$$

**Schedules.** Let

$$g_{t,k} := \|\nabla_w H(\theta_t, w_{t,k})\|_2, \qquad \gamma_{t,k}^{\text{safe}} = \frac{1}{g_{t,k}} q\left(\frac{g_{t,k}}{2}; g_{t,k}\right),$$

denote the local curvature-checked cap from Definition 4.5, and define the initialization-scale reference

$$\gamma_{0,0}^{\text{safe}} := \gamma^{\text{safe}}(\theta_0, \theta_0).$$

For the Poisson loss,

$$\gamma_{0,0}^{\text{safe}} \approx 0.04055,$$

while for the Skellam loss,

$$\gamma_{0,0}^{\text{safe}} \approx 0.02690.$$

We compare the curvature-checked schedule

$$\gamma_{t,k} = \min\{50\,\gamma_{0,0}^{\text{safe}},\ \gamma_{t,k}^{\text{safe}}\}$$

against four constant baselines,

$$\gamma_{t,k} \equiv c\,\gamma_{0,0}^{\text{safe}}, \qquad c \in \{0.5, 1, 5, 50\}.$$

This comparison is intentionally simple: all constant baselines are normalized by the same initial local cap, while the curvature-checked rule recomputes safety online.

**Diagnostics.** At each outer iteration we record:

$$\|\theta_t - \theta^\star\|_2, \qquad \|\nabla_w H(\theta_t, \theta_t)\|_2$$

where $\theta^\star$ is the fixed point solving

$$\nabla_w H(\theta^\star, \theta^\star) = 0.$$

We also log the actual learning rate used at each inner step and the corresponding local safe cap $\gamma_{t,k}^{\text{safe}}$.

**Results.** Figure 1 summarizes the outcomes.

Three observations are worth emphasizing.

First, for both losses, the curvature-checked rule remains stable and converges to the fixed point to numerical precision, even though its baseline parameter is set to the very aggressive value $50\,\gamma_{0,0}^{\text{safe}}$. This is possible because the actual update is clipped online to the local safe cap whenever the local curvature becomes too large.

Second, the constant-step baselines reveal the expected stability hierarchy. Conservative and moderately aggressive choices, such as $0.5\,\gamma_{0,0}^{\text{safe}}$, $1.0\,\gamma_{0,0}^{\text{safe}}$, and even $5.0\,\gamma_{0,0}^{\text{safe}}$, remain stable on this particular toy problem, although the smaller choices converge more slowly. In contrast, the highly aggressive constant choice $50\,\gamma_{0,0}^{\text{safe}}$ diverges almost immediately for both losses.

Third, the last-panel learning-rate plots make the curvature-check mechanism visible. For the curvature-checked run, the actual step

$$\gamma_{t,k} = \min\{50\,\gamma_{0,0}^{\text{safe}}, \gamma_{t,k}^{\text{safe}}\}$$

tracks the local safe cap whenever clipping is active, and returns to the high baseline level whenever the local geometry becomes benign again. In contrast, the fixed-step schedules ignore this variation and therefore cannot protect against sudden curvature amplification.

**Interpretation.** This experiment is intentionally modest. It is not intended to claim practical superiority over tuned fixed stepsizes in general RL. Rather, it illustrates the mechanism predicted by the theory in a theorem-aligned sandbox: once curvature can grow with the gradient scale, a constant stepsize chosen only from the initial geometry can become catastrophically unsafe, whereas the curvature-checked rule remains stable by adapting to the local profile. In this sense, the toy experiment provides a concrete visualization of the trajectory-dependent stability law developed in Sections 4–5.

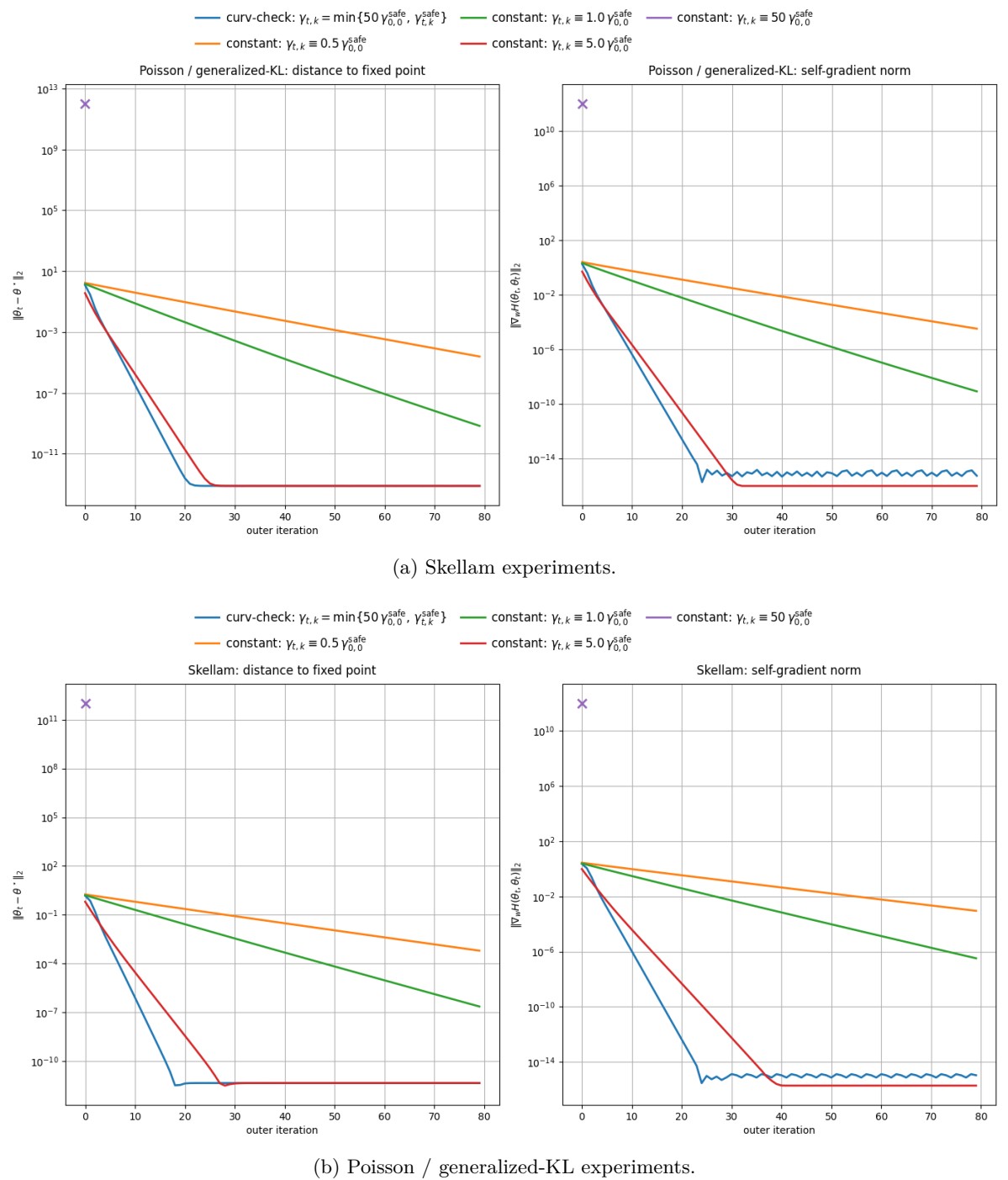

(a) Skellam experiments.

(b) Poisson / generalized-KL experiments.

Figure 1: Comparison of curvature-checked and constant-step schemes across two generalized smoothness examples. **Top:** Poisson / generalized-KL. **Bottom:** Skellam.

## A.4 Real-data experimental illustration

We next include a small real-data experiment in the same spirit as the toy example. The purpose is to examine the learning-rate stability pattern on a nontrivial feature space, while still keeping the construction controlled enough that the quantities appearing in the theory can be computed.

**Setup.** We build a simple latent Markov prediction problem from MNIST. The latent state is the digit label $s \in \{0, \ldots, 9\}$, and the observation is an MNIST image drawn from that class. To obtain features, we first train a small convolutional network on MNIST classification, then freeze the encoder and use its penultimate representation as the state feature map $\phi(x) \in \mathbb{R}^{32}$. The encoder has two convolutional layers with ReLU activations and $2 \times 2$ max-pooling, followed by two fully connected layers of widths 64 and 32. A final 10-class linear layer is used only during this pretraining stage. We train the network for 4 epochs with Adam, learning rate $3 \times 10^{-3}$, and batch size 256.

After freezing the encoder, we generate a dataset of $n = 20{,}000$ transitions as follows. We sample the current label $s$ uniformly from $\{0, \ldots, 9\}$. With probability 0.75, the next label is set to

$$s' = (s + 1) \pmod{10},$$

and with probability 0.25, $s'$ is drawn uniformly from $\{0, \ldots, 9\}$. We then sample an image $x$ from class $s$ and an image $x'$ from class $s'$, and define

$$\phi = \phi(x), \qquad \phi' = \phi(x').$$

The reward is

$$r = \mathbf{1}\{s = 9\}.$$

Thus, the experiment uses a 10-state latent environment with image observations and a sparse state-based reward. All feature coordinates are standardized before optimization.

**Loss.** For discount factor $\gamma = 0.95$, the TD target is

$$y_\theta = r + \gamma \, {\phi'}^\top \theta.$$

We use the symmetric exponential loss

$$H_{\cosh}(\theta, w) = \frac{1}{n} \sum_{i=1}^{n} \left( 2\cosh(\phi_i^\top w) - y_i(\theta) \, \phi_i^\top w \right) + \frac{\lambda}{2} \|w\|_2^2, \qquad \lambda = 5.$$

Equivalently, writing $\Phi, \Phi_{\text{next}} \in \mathbb{R}^{n \times d}$ for the matrices of current and next features,

$$H_{\cosh}(\theta, w) = \frac{1}{n} \sum_{i=1}^{n} \left( 2\cosh((\Phi w)_i) - y_i(\theta)(\Phi w)_i \right) + \frac{\lambda}{2} \|w\|_2^2.$$

In this experiment,

$$F_w = \lambda = 5, \qquad F_\theta = \gamma \left\| \frac{\Phi^\top \Phi_{\text{next}}}{n} \right\|_{\text{op}} = 3.578258,$$

so the force-dominance condition holds:

$$F_\theta < F_w.$$

**Schedules.** We initialize at $\theta_0 = 0$ and perform $K = 5$ inner gradient steps per outer iteration. For the clipped method, we use the $\theta$-dependent curvature envelope

$$\ell_\theta(s) = \lambda + \lambda_{\max}\left( \frac{\Phi^\top \Phi}{n} \right) 2\cosh\left( R_\phi \left( B_\theta + \frac{s}{\lambda} \right) \right), \qquad B_\theta := \|\theta\| + \frac{\|\nabla f_\theta(\theta)\|}{\lambda},$$

where

$$\lambda_{\max}\left( \frac{\Phi^\top \Phi}{n} \right) = 8.398357, \qquad R_\phi = \max_i \|\phi_i\|_2 = 9.340547.$$

The clipped update uses

$$\gamma_{t,k} = \min\{\gamma_0, \gamma_{\text{safe}}^{t,k}\},$$

while the fixed baseline uses

$$\gamma_{t,k} \equiv \gamma_0.$$

To ensure that clipping never activates, Theorem 5.1 requires the baseline stepsize to satisfy

$$\gamma_0 \leq \frac{1}{2\,\ell(3M/2)}.$$

However, this condition depends on the quantity $M$, which is trajectory-dependent and therefore not known in advance. For the experiments, we therefore use a relaxed, data-driven reference scale instead. Specifically, we first run a clipped trajectory for $(t,k) = (180,5)$ iterations with a deliberately large baseline stepsize

$$\gamma_0 = \frac{1.25}{F_w},$$

so that clipping is triggered throughout the run. Along this trajectory, we record the smallest safe stepsize encountered, namely

$$\hat{\gamma}_{\min} = 8.675228 \times 10^{-3}.$$

This quantity serves as a practical lower reference for the safe stepsize along the trajectory, and in particular satisfies

$$\gamma_{\min}^{\text{safe}} \leq \hat{\gamma}_{\min} \leq \frac{1}{F_w}.$$

We then scan both methods over the learning-rate values listed in Table 1. More precisely to see the effect of clipping, we bin the stepsize $\gamma_0$ into three regions as follows:

$$\gamma_0 \in \begin{cases} \underbrace{(0, \hat{\gamma}_{\min})}_{\text{low region}}, \\ \underbrace{\left[\hat{\gamma}_{\min}, \frac{1}{F_w}\right)}_{\text{medium region}}, \\ \underbrace{\left[\frac{1}{F_w}, \infty\right)}_{\text{high region}}. \end{cases}$$

Each scan run uses 160 outer iterations, as shown in Figure 2.

**Diagnostics.** At each outer iteration we record the Bellman residual

$$\frac{1}{n}\big\|\Phi\theta_t - (r + \gamma\Phi_{\text{next}}\theta_t)\big\|_2^2,$$

the self-gradient norm $\|\nabla_w H(\theta_t, \theta_t)\|_2$, the fraction of inner updates that are clipped (`clip frac.`), and the mean effective step used by the clipped rule (`mean used` $\gamma_0$).

**Results.** Table 1 summarizes the scan, and Figure 2 shows the gradient-norm trajectories across the learning-rate spectrum.

The same pattern as in the toy example appears clearly. For baselines below the observed threshold $\hat{\gamma}_{\min}$, the clipped rule is essentially inactive, and the fixed and clipped methods behave the same. As the baseline increases past this threshold, clipping activates but the method remains stable and reaches the same final Bellman residual. In contrast, sufficiently aggressive fixed baselines diverge, while the clipped rule remains stable for the same nominal values by reducing the actual step online. In the large-step regime, the clipping fraction is 1, and the mean effective step is approximately $2.19 \times 10^{-2}$ even when the nominal baseline is much larger. This is consistent with the stability picture developed in the theory: once curvature amplifies along the trajectory, a fixed step can become unsafe, whereas the clipped rule continues to adapt to the local scale.

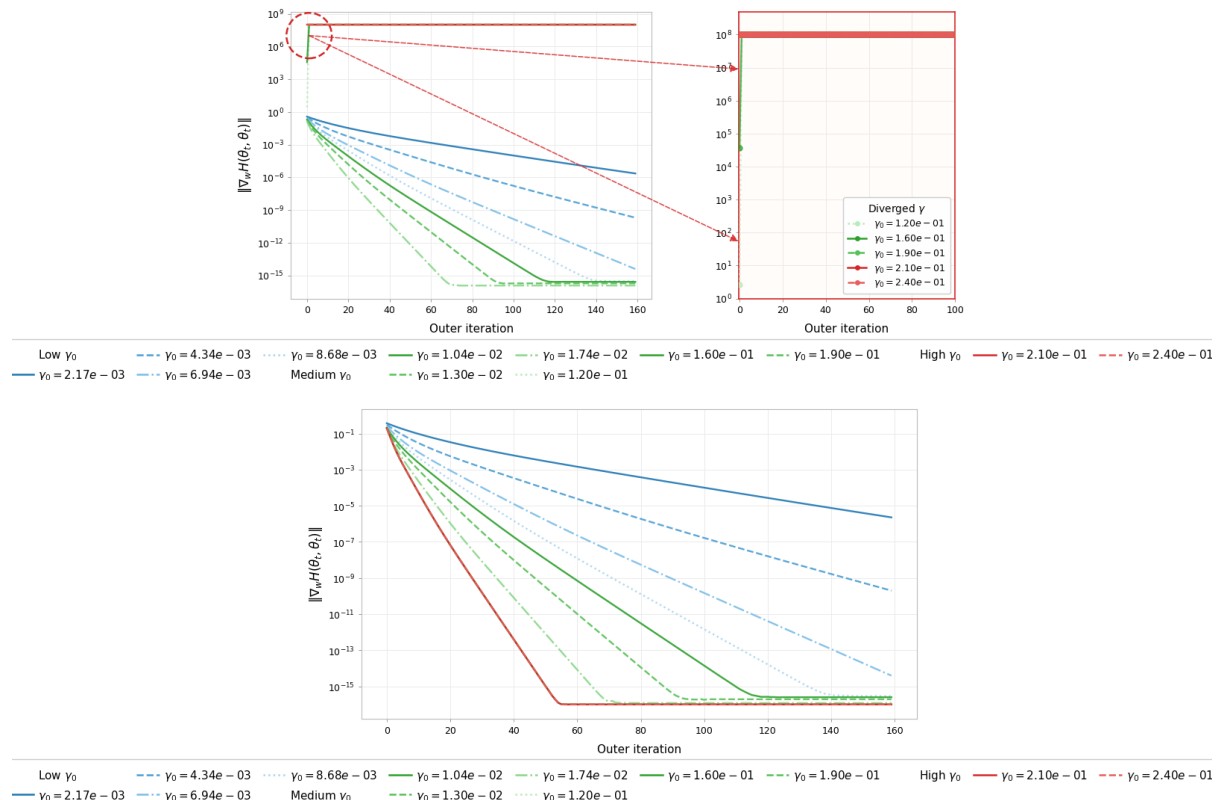

Figure 2: Gradient-norm trajectories across the scanned learning-rate spectrum. **Top:** fixed-step runs. Low baselines remain stable, while more aggressive baselines diverge. **Bottom:** clipped rule (10) remains stable across the same nominal baselines because the actual step adapts to the local safe scale, hence proving our theory.

Table 1: Summary of the real-data learning-rate scan for fixed and clipped updates.

| $\gamma_0$ | $\gamma_0 F_w$ | fixed div. | clip frac. | mean used $\gamma$ |
|---|---|---|---|---|
| 0.002169 | 0.010844 | ✗ | 0.00000 | 0.002169 |
| 0.004338 | 0.021688 | ✗ | 0.00000 | 0.004338 |
| 0.006940 | 0.034701 | ✗ | 0.00000 | 0.006940 |
| 0.008675 | 0.043376 | ✗ | 0.00000 | 0.008675 |
| 0.010410 | 0.052051 | ✗ | 0.00250 | 0.010407 |
| 0.013013 | 0.065064 | ✗ | 0.00625 | 0.012998 |
| 0.017350 | 0.086752 | ✗ | 0.00750 | 0.017308 |
| 0.120000 | 0.600000 | ✓ | 1.00000 | 0.022114 |
| 0.160000 | 0.800000 | ✓ | 1.00000 | 0.022114 |
| 0.190000 | 0.950000 | ✓ | 1.00000 | 0.022114 |
| 0.210000 | 1.050000 | ✓ | 1.00000 | 0.022114 |
| 0.240000 | 1.200000 | ✓ | 1.00000 | 0.022114 |

