# OpenReview forum: "Global Linear Convergence of Inexact TD Under Generalized Smoothness"
_TMLR — Under review for TMLR_

### Review · Reviewer_WH3n · 2026-03-23

**Summary Of Contributions:**

This paper studies the optimization view and convergence of temporal-difference learning with target networks. The main contribution is the elimination of the global smoothness assumption used in prior work, while retaining the same force-dominance results from prior work (concretely, going from global 𝐿-smoothness to gradient-scale-dependent curvature growth). This makes the result more realistic from an optimization perspective, even though it still stops short of the fully stochastic, nonconvex deep-RL setting.

**Strengths**

- The paper is very well written, clearly establishes the background and problem setting, and is well-situated in terms of related-work.
- The paper clearly provides all necessary background tools needed to understand their analysis (section 3)
- The authors identify a precise limitation in prior work (reliance on global 𝐿-smoothness) and replace it with a more general curvature-growth model. This is a meaningful and well-scoped extension of the force-dominance analysis of  Asadi et al. (2023b).
- The paper explicitly recovers the uniformly smooth regime as a special case.

**Weaknesses**

- Assumptions 4.1 and 4.2 are still quite strong from a practical RL point of view. This is a bit disconnected from RL in any meaningfully complex task and the practical implications for modern TD with neural networks remain indirect.
- If I understand correctly, ℓ(.) is a nondecreasing positive function that upper-bounds how large the Hessian norm can be as a function of the current gradient magnitude. The paper’s step-size rule depends on ℓ directly. In the limitations section, the authors mention that would need to be estimated online in practice. However, it is unclear how unclear how ℓ would be specified, learned, or even approximated (especially for neural network approximators).
- The paper currently has no experiments. While I am not aware of the experimental expectations in this community, I think that a lack of empirical analysis makes the contribution hard to verify. Is it possible to verify some of your results through small experiments, perhaps in discrete settings with simple approximators?
- In the conclusions, the authors claim that their work is a theoretical bridge between the empirical characteristics modern deep RL and classical optimization. I find this claim to be a bit overstated. As mentioned above, the paper has no empirical validation in deep RL settings (or even in simple grid-world domains). Further, it is unclear whether Assumptions 4.1 and 4.2 are realistic in the deep RL setting or if quantities like M or ℓ(.) can be estimated online.

**Questions**

- Can you give a concrete TD-relevant example where generalized smoothness holds but global smoothness fails?

**Audience:**

Yes

**Audience Explanation:**

The paper contributes to the fields of theoretical RL and optimization by presenting an extension of prior results and an improvement in terms of some assumptions. I believe that TMLR's audience would find this paper interesting and fitting the journal's scope.

**Claims And Evidence:**

Yes

**Claims Explanation:**

The paper claims several theoretical advancements and supports them via proofs and mathematical reasoning. I find most claims to be well-supported, except for the broader applicability to deep RL. I kindly suggest the authors to rewrite this claim to make it clear that the connection to deep RL is limited owing to assumptions like lipschitzness and convexity.

**Requested Changes:**

- Is it possible to verify your results through small experiments, perhaps in discrete settings with simple approximators?
- Please tone down the claim about applicability to deep RL (or support this through experiments or a longer discussion on what assumptions fail and what properties can still be retained).

---

> ### Author Response · Authors · 2026-04-06
> **Response to Reviewer WH3n (1)**
>
> We thank the reviewer for the thoughtful and detailed feedback. We are encouraged that the reviewer views the paper a relevant, and potentially valuable to the TMLR audience. Below, we address the main concerns point by point and outline the concrete changes we have made in the revision. We first start by answering the question:
>
> $\textbf{Question by WH3n:}$ Can you give a concrete TD-relevant example where generalized smoothness holds but global smoothness fails?
>
> $\textbf{Response:}$  We thank the reviewer for this helpful suggestion. In the revision, we added a dedicated appendix subsection (Appendix A.2) precisely to address this point, in the same spirit as the illustrative examples in Asadi et al. (2023, Sec. 6.3). $\textit{The additional sections make sure that the domain covered by our assumptions is non-empty and concrete, rather than merely formal.}$
>
> Our examples serve as concrete TD-structured witnesses showing that our assumption class is nonempty once one adopts the stabilized inner-problem viewpoint already used in the optimization perspective of TD. This is a natural request in the present setting because RL already uses non-quadratic Bellman-style objectives, for example, the logistic Bellman error in Q-REPS [1] and Huber-type losses in distributional RL [2]. The question is therefore not whether one can move beyond the quadratic loss, but whether one can do so while retaining a clean TD structure and allowing genuinely non-uniform upper curvature.
>
> Concretely, we now provide two explicit TD-structured examples where the generalized-smoothness assumptions can be verified in closed form: $\textbf{(i)}$ a Poisson / generalized-KL Bellman loss [3,4], and $\textbf{(ii)}$ a Skellam / cosh Bellman loss [5,6]. For both examples, after ridge stabilization, Assumption 4.2 holds globally with $F_w=\lambda$, and Assumption 4.1 follows from a Lipschitz target map with $F_\theta$.
>
> More importantly, for each frozen inner problem $f_t(w)=H(\theta_t,w)$, we derive explicit local curvature-growth profiles $\ell_t(\cdot)$ for both Poission and Skellman cases (explicit equations can be found in the Appendix A.2). Thus, generalized smoothness holds while global smoothness fails, since the upper curvature grows with the gradient/target scale. We then show how the bounded-iterate result (Lemma 4.7) yields uniform envelopes $\bar{\ell}(\cdot)$, allowing Theorem 5.1 to apply directly and making the implications of the theory explicit: the admissible constant baseline stepsize becomes $\gamma_0 \le [2\bar{\ell}(3M/2)]^{-1}$, and the inner effort required for a fixed-strength contraction scales as $K \gtrsim \bar{\ell}(3M/2)/\lambda$. In other words, these examples make concrete the paper’s main message: compared with the globally smooth setting, curvature amplification along the trajectory tightens the safe constant-step threshold and increases the inner optimization effort needed for stability. These examples were added specifically to demonstrate that the paper’s assumption class has concrete witnesses and that the resulting stability law has a clear interpretation in theorem-aligned TD settings.
>
> The main purpose of Appendix A.2 is therefore not only illustrative but existential: it shows that the class of problems covered by our theory is genuinely populated.
>
> Reference:
>
> [1] Joan Bas-Serrano, Sebastian Curi, Andreas Krause, and Gergely Neu. "Logistic q-learning". In International
> conference on artificial intelligence and statistics, pp. 3610–3618. PMLR, 2021.
>
> [2] Will Dabney, Mark Rowland, Marc Bellemare, and Rémi Munos. "Distributional reinforcement learning with
> quantile regression". In Proceedings of the AAAI conference on artificial intelligence, volume 32, 2018.
>
> [3] John Ashworth Nelder and Robert WM Wedderburn. "Generalized linear models". Journal of the Royal
> Statistical Society Series A: Statistics in Society, 135(3):370–384, 1972.
>
> [4] Daniel Lee and H Sebastian Seung. "Algorithms for non-negative matrix factorization". Advances in neural
> information processing systems, 13, 2000.
>
> [5] Siem Jan Koopman, Rutger Lit, and André Lucas. "The dynamic skellam model with applications". 2014.
>
> [6] Konstantinos Pelechrinis and Wayne Winston. "A skellam regression model for quantifying positional value
> in soccer". arXiv preprint arXiv:1807.07536, 2018.

---

> ### Author Response · Authors · 2026-04-06
> **Response to Reviewer WH3n (2)**
>
> $\textbf{Q:}$ If I understand correctly, $\ell(\cdot)$ is a nondecreasing positive function that upper-bounds how large the Hessian norm can be as a function of the current gradient magnitude. The paper’s step-size rule depends on $\ell(\cdot)$ directly. In the limitations section, the authors mention that would need to be estimated online in practice. However, it is unclear how unclear how $\ell(\cdot)$ would be specified, learned, or even approximated (especially for neural network approximators).
>
> $\textbf{Resp:}$ On the quantities $M$ and $\ell(\cdot)$, we agree that practical online estimation is an important issue, but it is distinct from the goal of the present paper. This work is a convergence-analysis paper: it characterizes what follows if a valid curvature-growth envelope is available. That is standard in optimization theory; classical results are also routinely stated in terms of smoothness/Lipschitz quantities or condition numbers that are not known exactly a priori and are not themselves estimated by the theorem (especially stepsize are bounded in terms of unknown Lipschitz constants) [1,2,3]. In our setting, $M$ is used as an analysis device to express a uniform admissibility law for a constant baseline step; it is not an online input required by the algorithm.
>
> $\textbf{Estimation of}  \ell(\cdot):$ The quantity that needs to be approximated for practical deployment is the local curvature profile ℓ(⋅), and we now explicitly frame the robust online estimation of such profiles for generic neural critics as an important direction for future work. In our framework, $\ell(\cdot)$ is a relaxation of the classical global Lipschitz-smoothness assumption, since the uniformly smooth case is recovered by taking $\ell(\cdot) = L$. There is already precedent for online estimation of regularity surrogates in simpler settings: AdaLIPO [4] estimates a global Lipschitz constant during the optimization process, and LACKI [5] develops an online estimator of a Hölder constant from incrementally arriving noisy data, with explicit motivation toward settings where the regularity constant is not available a priori. We cite these works not to claim that they already solve our problem, but to indicate that the broader program of estimating regularity information online is viable and well-motivated. These works show that online estimation of regularity information is possible in simpler global settings. However, our setting is materially harder: we require estimation not of a single global constant, but of a nondecreasing curvature-growth profile $\ell(\cdot)$ suitable for nonstationary TD objectives with neural critics. Developing such estimators, together with finite-sample guarantees and numerically stable implementations, would constitute a substantial project in its own right. By contrast, designing statistically and computationally robust procedures for estimating a state-dependent/local curvature profile $\ell(\cdot)$ for high-capacity neural critics is, in our view, a substantial new research direction and lies beyond the scope of the present paper.
>
> $\textbf{Our take in estimating}  \ell(\cdot):$ At the same time, we do not leave $\ell(\cdot)$ entirely abstract: In Appendix A.2, we go further and make the generalized-smoothness profile concrete in analytically tractable TD-style examples: for simple stabilized losses, we derive explicit nondecreasing curvature profiles $\ell(\cdot)$, show that the curvature-checked stepsize can be instantiated directly from these profiles, and then upper-bound them by time-independent envelopes so that Theorem 5.1 applies with fully explicit admissible stepsize and inner-effort scaling laws. Hence, Appendix A.2 already provides a constructive route for specifying or upper-bounding $\ell(\cdot)$ in simple cases, rather than treating it as a purely abstract assumption. What we do not solve in this paper is the substantially harder problem of designing a robust, generic online estimator for a state-dependent curvature profile for high-capacity neural critics; we believe that problem is important, but it is a separate research direction beyond the scope of the present theoretical contribution
>
> We therefore view the appropriate claim as follows: the present paper establishes a conditional convergence theory under generalized smoothness in a stabilized TD regime, while the design of practically robust online estimators
>
> References:
>
> [1] Li et.al, "Convex and Non-convex Optimization Under Generalized Smoothness". NeurIPS 2023.
>
> [2] Chezhegov et.al, "Clipping Improves Adam-Norm and AdaGrad-Norm when the Noise Is Heavy-Tailed", ICML 2025.
>
> [3] Li et.al, "A Unified Analysis of Stochastic Gradient Descent with Arbitrary Data Permutations and Beyond", NeurIPS 2025.
>
> [4] Malherbe et.al, "Global optimization of Lipschitz functions", ICML 2017.
>
> [5] Calliess et.al, "Lazily Adapted Constant Kinky Inference for Nonparametric Regression and Model-Reference Adaptive Control", Automatica 2020.

---

> ### Author Response · Authors · 2026-04-06
> **Response to Reviewer WH3n (3)**
>
> $\textbf{Q:}$ The paper currently has no experiments. While I am not aware of the experimental expectations in this community, I think that a lack of empirical analysis makes the contribution hard to verify. Is it possible to verify some of your results through small experiments, perhaps in discrete settings with simple approximators.
>
> $\textbf{Resp:}$ We added small experiments in the Appendix. Most directly, Appendix A.3 now includes a deterministic 4-state prediction problem with a simple direct-coordinate parameterization, using the explicit TD-structured losses introduced in Appendix A.2 (Poisson / generalized-KL and Skellam / cosh Bellman losses). These losses are chosen because they satisfy the force dominance condition and have a nondecreasing Hessian upper profile, which exactly matches our analysis. In this setting, the relevant quantities in the theory are computable $(F_\theta, F_w, \ell() function)$, the force-dominance condition can be checked explicitly, and we compare the proposed curvature-checked rule against fixed constant-step baselines normalized by the same initial safe scale. The resulting behavior matches the theory: conservative and moderately aggressive fixed steps remain stable, highly aggressive fixed steps diverge almost immediately, while the curvature-checked rule remains stable and converges by clipping to the local safe scale when curvature amplifies.
>
> We also include a learned-feature real-data illustration in Appendix A.4 that shows the same stability pattern in a nontrivial feature space. We emphasize that these are theorem-aligned experiments rather than claims about broad deep-RL performance, but they directly verify the stability mechanism our analysis is designed to capture.

---

> ### Author Response · Authors · 2026-04-06
> **Response to Reviewer WH3n (4)**
>
> $\textbf{Q:} [Requested Changes]$ Please tone down the claim about applicability to deep RL (or support this through experiments or a longer discussion on what assumptions fail and what properties can still be retained).
>
> $\textbf{Resp:}$ We thank the reviewer for pointing this out. We agree that our earlier phrasing overstated the connection to modern deep RL. In the revision, we have toned down the relevant statements throughout the paper and clarified the intended scope: our results give a mechanism-level analysis within a stabilized optimization view of TD, rather than a full convergence theory for stochastic nonconvex deep RL. Concretely, we now describe the paper as isolating the effect of replacing global smoothness with trajectory-dependent curvature growth while retaining the force-dominance framework $(F_\theta < F_w)$, and we avoid language that suggests direct applicability to generic stochastic nonconvex neural-TD settings. All the changes in the revised manuscript are written in purple colour.
>
> Specifically, we have made the following changes in our draft to tone down the overclaim:
>
> | S. No. | What part of the paper it is | In the Submitted Manuscript | In the Revised Manuscript | Justification |
> |---|---|---|---|---|
> | 1 | Conclusion, final sentence on pages 12--13 | Ultimately, this provides a rigorous theoretical bridge between the empirical pathologies of modern deep RL and classical optimization, demonstrating that robust stepsize fail-safes and fast geometric contraction are not mutually exclusive. | **Ultimately, our results give a rigorous analysis of trajectory-dependent curvature amplification within a stabilized optimization view of TD, showing that robust stepsize fail-safes and fast geometric contraction are not mutually exclusive in that setting.** | It removes the unsupported “bridge to modern deep RL.” |
> | 2 | Abstract, last sentence on page 1 | In the uniformly smooth case, we recover Asadi et al.\ (2023b), while under curvature growth, the worst trajectory gradient scale controls both stability and attainable convergence speed, aligning with step-control heuristics used in reinforcement learning (RL). | **In the uniformly smooth case, we recover Asadi et al.\ (2023b), while under curvature growth, the worst trajectory gradient scale controls both stability and attainable convergence speed, yielding a mechanism-level interpretation of why curvature-aware step control can matter in stabilized TD-style optimization.** | This keeps the motivation, but removes the broader RL claim. |
> | 3 | Introduction, deep-RL motivation paragraph on page 2 | In modern deep RL with neural critics and value heads, the inner optimization geometry can exhibit pronounced, highly non-uniform upper curvature. | **As motivation, modern deep RL with neural critics and value heads can exhibit pronounced, highly non-uniform upper curvature in the associated optimization landscapes.** | This is close to claiming that our analysis directly covers that regime. We have now explicitly framed it as a source of inspiration/motivation, not as the setting of the theorem. |
> | 4 | Same paragraph, sentence on practical prediction | In such regimes, a single global $L$ controlling $\|\|\nabla_w^2 H(\theta,w)\|\|_{\mathrm{op}}$ is either unavailable or so conservative that it ceases to predict the effective stepsizes that remain stable in practice, even for stabilized or regularized inner objectives. | **In such regimes, a single global $L$ controlling $\|\|\nabla_w^2 H(\theta,w)\|\|_{\mathrm{op}}$ may be unavailable or overly conservative as a descriptor of the locally stable stepsizes encountered along training trajectories, even for stabilized or regularized inner objectives.** | Earlier, this read as a fairly strong empirical claim about practice. Now, we have it only observationally and in a limited form. |
> | 5 | The scope paragraph on page 2: strengthen it slightly | This modeling choice is used only to remove nonconvex optimization pathologies; the phenomenon we isolate is upper-curvature amplification along the trajectory. | **We have added a sentence following this, “Accordingly, our results should be read as a mechanism-level analysis within a stabilized optimization view of TD, rather than as a full convergence theory for stochastic nonconvex deep RL.”** | This sentence is added to clearly highlight that we do not claim to provide a full convergence theory for deep nonconvex RL; we only claim to provide convergence guarantees under a stabilized force-dominance regime with non-uniformly bounded Hessians. |
> | 6 | Section 2.1, practical-ramifications sentence on page 5 | It also has important practical ramifications when designing RL optimizers (Asadi et al., 2023a). | **It also helps motivate optimizer design questions in RL (Asadi et al., 2023a).** | This is minor, but it still sounds stronger than necessary. |

---

> ### Author Response · Authors · 2026-04-06
> **Response to Reviewer WH3n (5)**
>
> $\textbf{Q:}$ Assumptions 4.1 and 4.2 are still quite strong from a practical RL point of view. This is a bit disconnected from RL in any meaningfully complex task and the practical implications for modern TD with neural networks remain indirect.
>
> $\textbf{Resp:}$ We agree that Assumptions 4.1 and 4.2 are strong from the standpoint of practical deep RL, and we have revised the paper to make this scope explicit. Our aim is not to provide a full convergence theory for stochastic, nonconvex TD with generic neural critics. Rather, we deliberately work within the stabilized optimization-view regime already used in prior force-dominance analyses by Asadi et. al, 2023, so that we can isolate a different bottleneck left unresolved there: the reliance on global smoothness to certify inner descent. Within that scope, our contribution is to replace the global 𝐿-smoothness assumption by trajectory-dependent curvature growth, while retaining the same force-dominance mechanism and making the resulting stability law explicit. In the revised manuscript, we now state this more directly in the introduction and theory sections, emphasizing that the result is a mechanism-level analysis within a stabilized TD setting rather than a direct theorem for generic deep-RL critics (changes are written in purple colour). The practical implication is therefore indirect but precise: the results characterize how curvature amplification changes admissible constant stepsizes and the inner effort required for stable TD-style optimization once a stabilized local model is available.
>
> In the above cell, Response (4), we have explicitly stated our changes to tone down the earlier claims.

---

> > ### Comment · Reviewer_WH3n · 2026-05-07
> > **Rebuttal Acknowledgement**
> >
> > I thank the authors for their responses and revisions. Most of my concerns have been resolved. As a minor point, I encourage the authors to consider including an illustration of their theoretical mechanism. This would greatly help readers understand the background and abstract theoretical concepts that the paper expands on.

---

> ### Author Response · Authors · 2026-05-11
> **Reply to reviewer WH3n**
>
> We thank the reviewer for this helpful suggestion. We agree that a concrete illustration of the theoretical mechanism can make the abstract generalized-smoothness and stability concepts easier to follow.
>
> In this direction, we would respectfully like to highlight the illustrative material that is already included in the manuscript:
>
> $\textbf{1. Constructive examples in Appendix A.2.}$
>
> These examples instantiate our assumptions in explicit TD-structured settings. They show that the problem class considered in the paper is non-vacuous and help clarify how the generalized-smoothness condition leads to a concrete stability law.
>
> $\textbf{2. Mechanism-oriented illustrations in Appendices A.3 and A.4.}$
>
>  These toy and real-data experiments provide a sanity check for the theoretical mechanism. They illustrate the contrast between aggressive fixed stepsizes, which can become unstable, and the curvature-checked rule, which remains stable by adapting to the local safe scale.
>
> We hope these examples help address the reviewer’s suggestion and clarify the intuition behind the theoretical results. We also appreciate the suggestion of an illustration, and would be happy to add an additional schematic figure in a future version if the reviewer feels that it would further improve readability.

---

### Review · Reviewer_PGFA · 2026-03-30

**Summary Of Contributions:**

This paper provides a theoretical analysis of Temporal-Difference (TD) learning with target networks, framing it as a two-level iterative optimization process. In this framework, the inner problem involves a frozen target parameter \theta that is not solved exactly for the optimization parameter w. While previous work (Asadi et al., 2023b) established stability under a "force-dominance" condition, their analysis relied on global L-smoothness (a uniform curvature bound). This paper generalizes those results by:

_Removing Global Smoothness: It extends prior work on TD convergence by removing the uniform bound on the Hessian of the objective, replacing it with a "generalized smoothness" model where the Hessian magnitude grows with the local gradient scale.

_Curvature-Checked Stepsize Rule: It proposes a new stepsize rule using a safety cap derived from the curvature-growth profile to ensure inner optimization descent without requiring prior knowledge of global curvature constants.

_Stability Scaling Law: It identifies a relationship where the largest admissible stepsize decays and the required number of inner steps K increases proportionally to the curvature encountered during training.

Strengths:

• Addresses the "edge-of-stability" behavior observed in deep RL.

• Provides a rigorous framework that maintains the well-posedness of the inner problem (strong convexity) while isolating the effects of unbounded upper curvature.

Weaknesses:

• Language Accuracy: The introduction states that the impact of the work "extends well beyond RL to areas such as AI." This is an inexact sentence as RL is a subfield of AI and not a separate area.

• Non-standard Abstract Formatting: The abstract contains formal citations (Asadi et al., 2023b). It is standard academic practice to make the abstract self-contained. Formal citations should generally be avoided in this section.

• Lack of Examples: The paper would benefit from providing a few examples (similar to Section 6.3 of Asadi et al., 2023) where the specific assumptions of this paper hold. Fully describing the work under those examples would add value to the current work and improve its clarity.

**Audience:**

Yes

**Audience Explanation:**

The paper addresses fundamental convergence questions for TD learning. By relaxing the restrictive global smoothness assumption, the work aligns theoretical analysis with the empirical behavior of neural network training in AI, such as non-uniform curvature. Researchers focused on the stability of value-based RL algorithms would find these useful.

**Broader Impact Concerns:**

No concerns.

**Claims And Evidence:**

Yes

**Claims Explanation:**

The authors provide a detailed mathematical framework extending prior force-dominance theory to the generalized smoothness setting. They derive a one-step descent certificate (Lemma 4.6) that does not rely on global Lipschitz gradients. This leads to an inner-step contraction (Proposition 3) and a cumulative outer recursion (Proposition 4) that yields a global linear convergence rate. The evidence is structured through clear assumptions and technical lemmas.

**Requested Changes:**

Critical changes (required for acceptance):
_Correct the language in the introduction regarding the relationship between RL and AI to reflect that RL is a subfield of AI.

Strengthening the work:

_Provide specific examples (similar to Asadi et al. 2023, Section 6.3) where the generalized smoothness assumptions hold. Describe the implications of the theory within the context of these examples to improve clarity.

_Notation: While the use of \theta and w follows the convention of recent optimization-centric RL literature (e.g., Asadi et al. 2023), the authors might consider a brief note acknowledging the more standard RL notation (using a single variable with a minus superscript for the target, \theta^-) to improve readability for a broader RL audience.

---

> ### Author Response · Authors · 2026-04-06
> **Response to Reviewer PGFA (1)**
>
> We thank the reviewer for the thoughtful and detailed feedback. We are encouraged that the reviewer views the paper a relevant, and potentially valuable to the TMLR audience. Below, we address the main concerns point by point and outline the concrete changes we have made in the revision. All revisions in the updated manuscript are written in red colour.
>
> $\textbf{ Critical changes (required for acceptance):}$ _Correct the language in the introduction regarding the relationship between RL and AI to reflect that RL is a subfield of AI.
>
> $\textbf{Resp:}$ We thank the reviewer for pointing this out. We agree that the earlier phrasing was imprecise, since RL is a subfield of AI. We have revised the introduction to state that TD has influence beyond RL itself, including in other areas of AI, as well as in economics and neuroscience.
>
> Kindly let us know if we need to make any other changes.
>
> $\textbf{Q:}$  _Provide specific examples (similar to Asadi et al. 2023, Section 6.3) where the generalized smoothness assumptions hold. Describe the implications of the theory within the context of these examples to improve clarity.
>
> $\textbf{Resp:}$  We thank the reviewer for this helpful suggestion. In the revision, we added a dedicated appendix subsection (Appendix A.2) precisely to address this point, in the same spirit as the illustrative examples in Asadi et al. (2023, Sec. 6.3). $\textit{The additional sections make sure that the domain covered by our assumptions is non-empty and concrete, rather than merely formal.}$
>
> Our examples serve as concrete TD-structured witnesses showing that our assumption class is nonempty once one adopts the stabilized inner-problem viewpoint already used in the optimization perspective of TD. This is a natural request in the present setting because RL already uses non-quadratic Bellman-style objectives, for example, the logistic Bellman error in Q-REPS [1] and Huber-type losses in distributional RL [2]. The question is therefore not whether one can move beyond the quadratic loss, but whether one can do so while retaining a clean TD structure and allowing genuinely non-uniform upper curvature.
>
> Concretely, we now provide two explicit TD-structured examples where the generalized-smoothness assumptions can be verified in closed form: $\textbf{(i)}$ a Poisson / generalized-KL Bellman loss [3,4], and $\textbf{(ii)}$ a Skellam / cosh Bellman loss [5,6]. For both examples, after ridge stabilization, Assumption 4.2 holds globally with $F_w=\lambda$, and Assumption 4.1 follows from a Lipschitz target map with $F_\theta$.
>
> More importantly, for each frozen inner problem $f_t(w)=H(\theta_t,w)$, we derive explicit local curvature-growth profiles $\ell_t(\cdot)$ for both Poission and Skellman cases (explicit equations can be found in the Appendix A.2). Thus, generalized smoothness holds while global smoothness fails, since the upper curvature grows with the gradient/target scale. We then show how the bounded-iterate result (Lemma 4.7) yields uniform envelopes $\bar{\ell}(\cdot)$, allowing Theorem 5.1 to apply directly and making the implications of the theory explicit: the admissible constant baseline stepsize becomes $\gamma_0 \le [2\bar{\ell}(3M/2)]^{-1}$, and the inner effort required for a fixed-strength contraction scales as $K \gtrsim \bar{\ell}(3M/2)/\lambda$. In other words, these examples make concrete the paper’s main message: compared with the globally smooth setting, curvature amplification along the trajectory tightens the safe constant-step threshold and increases the inner optimization effort needed for stability. These examples were added specifically to demonstrate that the paper’s assumption class has concrete witnesses and that the resulting stability law has a clear interpretation in theorem-aligned TD settings. The main purpose of Appendix A.2 is therefore not only illustrative but existential: it shows that the class of problems covered by our theory is genuinely populated.
>
> Ref:
>
> [1] Joan Bas-Serrano, Sebastian Curi, Andreas Krause, and Gergely Neu. "Logistic q-learning". In International
> conference on artificial intelligence and statistics, pp. 3610–3618. PMLR, 2021.
>
> [2] Will Dabney, Mark Rowland, Marc Bellemare, and Rémi Munos. "Distributional reinforcement learning with
> quantile regression". In Proceedings of the AAAI conference on artificial intelligence, volume 32, 2018.
>
> [3] John Ashworth Nelder and Robert WM Wedderburn. "Generalized linear models". Journal of the Royal
> Statistical Society Series A: Statistics in Society, 135(3):370–384, 1972.
>
> [4] Daniel Lee and H Sebastian Seung. "Algorithms for non-negative matrix factorization". Advances in neural
> information processing systems, 13, 2000.
>
> [5] Siem Jan Koopman, Rutger Lit, and André Lucas. "The dynamic skellam model with applications". 2014.
>
> [6] Konstantinos Pelechrinis and Wayne Winston. "A skellam regression model for quantifying positional value
> in soccer". arXiv preprint arXiv:1807.07536, 2018.

---

> ### Author Response · Authors · 2026-04-06
> **Response to Reviewer PGFA (2)**
>
> $\textbf{Notation:}$ While the use of \theta and w follows the convention of recent optimization-centric RL literature (e.g., Asadi et al. 2023), the authors might consider a brief note acknowledging the more standard RL notation (using a single variable with a minus superscript for the target, \theta^-) to improve readability for a broader RL audience.
>
> $\textbf{Resp:}$ Thank you for pointing this out. To improve the readability for a border RL audience, we have added the following footnote $\textit{“While we adopt $\theta$ to denote the target variable, consistent with recent optimization-centric RL literature (Asadi et.al, 2023), we wish to clarify for the broader RL community that this variable plays the functional role of the traditional target network weights, often denoted as $\theta^-$”}$ in at page 1 in Eq. (1) of the revised manuscript. This is written in red colour in the updated manuscript.
>
> Kindly let us know if we need to add anything else. Thank you.

---

> ### Author Response · Authors · 2026-04-06
> **Response to Reviewer PFGA (3)**
>
> $\textbf{ Non-standard Abstract Formatting:}$ The abstract contains formal citations (Asadi et al., 2023b). It is standard academic practice to make the abstract self-contained. Formal citations should generally be avoided in this section.
>
> $\textbf{Resp:}$ We thank the reviewer for pointing this out. We have now updated our abstract and made it self-contained. Kindly check our revised abstract in the revised manuscript (written in red colour).